# Robust Equation Structure Learning with Adaptive Refinement

**Yunlun Li**    **Sinno Jialin Pan**
Department of Computer Science and Engineering
The Chinese University of Hong Kong, Hong Kong
`yunlun.li@link.cuhk.edu.hk, sinnopan@cuhk.edu.hk`

## Abstract

Symbolic regression (SR) aims to automate scientific discovery, but often truncates the hypothetico–deductive cycle, focusing on hypothesis and experiment while lacking systematic analysis. We introduce RESTART, a framework that closes this loop by adding a principled analysis stage to diagnose and correct structural errors. RESTART features two core mechanisms: a short-term refinement process that uses boosting to identify unexplained signals and guide an LLM toward targeted corrections, and a long-term structure library that distills successful refinements into reusable code snippets for cumulative knowledge. On LLM-SRBench across Physics, Biology, and Materials Science, RESTART achieves lower error and higher accuracy than state-of-the-art baselines. It also generalizes robustly, recovering near-exact functional forms on out-of-distribution data, representing a significant advance toward fully automated scientific discovery.

## 1 Introduction

Scientific progress often unfolds through a simple yet profound *hypothesize–experiment–analyze* loop (bottom left of Figure 1): a scientist proposes a model, tests it against observations, identifies its shortcomings, and refines the hypothesis for next iteration. This iterative cycle powered discoveries from Kepler's laws of planetary motion to Newton's formulation of classical mechanics, and remains the foundation of modern science (Nola & Sankey, 2014; Li et al., 2021). Automating this process is the goal of *symbolic regression* (SR), which seeks to recover human-interpretable mathematical expressions from data. By discovering equations that both fit observations and remain human-readable, SR has the potential to accelerate scientific insight across domains ranging from physics to biology (Sun et al., 2023; Cranmer et al., 2020; Shi et al., 2023). However, SR is NP-hard (Udrescu & Tegmark, 2020), as the search space of expressions grows combinatorially with expression length and operator set, making efficient exploration critical.

Existing SR methods can be broadly categorized into *search-based* and *mapping-based* approaches (Shojaee et al., 2025a). Search-based methods such as genetic programming (GP) (Koza & Poli, 2005; Dubčáková, 2011; Cranmer, 2023; Mundhenk et al., 2021) evolve candidate populations via mutation and crossover, but rely heavily on random exploration and often revisit similar regions of the search space, leading to slow convergence. Mapping-based methods employ autoregressive models such as Transformers (Vaswani et al., 2017; Kamienny et al., 2023; Biggio et al., 2021) trained on large synthetic datasets to directly map numerical data to symbolic expressions. They can produce strong single-shot hypotheses, but lack an explicit mechanism for refining hypotheses based on observed error patterns, making them brittle in out-of-distribution (OOD) settings. Recent work has begun to incorporate Large Language Models (LLMs) into SR, leveraging their capacity for symbolic reasoning and natural language priors (Shojaee et al., 2025a; Grayeli et al., 2024; Ma et al., 2024; Wang et al., 2025). However, most such methods still instantiate only the first two steps of the scientific loop—hypothesize and experiment—without a principled analysis mechanism that turns observed errors into explicit guidance for iterative refinement.

We introduce **RESTART**[1] (**R**obust **E**quation **ST**ructure learning with **A**daptive **R**efinemen**T**), a novel framework that explicitly closes the hypothesize–experiment–analyze loop for SR. RESTART

---

[1]The code can be found at https://github.com/Liyunlun/RESTART.

starts with a mapping-based initializer to generate a strong first hypothesis, then uses an LLM to iteratively refine it under a two-level guidance mechanism as follows:

- **Short-term Guidance (Targeted Refinement):** After each test, we learn an exploration function to model what the currently learned equation fails to explain, providing highly specific, localized feedback to steer the next hypothesis.
- **Long-term Guidance (Cumulative Knowledge):** We maintain a persistent structure library to store validated refinements. Using an improvement-gated admission policy, we ensure only performance-improving refinements are retained, preventing library bloat and enabling efficient knowledge reuse.

Together, these components form a complete scientific cycle, emulating how human scientists iteratively refine their models. The empirical results show that our method converges faster, and discovers more accurate and parsimonious equations. Our **contributions** are summarized as follows:

- We propose RESTART, a framework that operationalizes the entire scientific discovery cycle. Inspired by how human scientists refine hypotheses, RESTART employs a short-term guidance mechanism to learn a data-driven exploration function for localized refinement. Concurrently, a long-term structure library cumulatively stores only performance-improving refinements, enabling reusable knowledge across iterations.
- Extensive empirical studies on LLM-SRbench (Shojaee et al., 2025b) show that RESTART achieves lower error and higher recovery accuracy than state-of-the-art baselines, including GP-based, mapping-based, and LLM-based methods.

## 2    PROBLEM FORMULATION

**Symbolic Regression (SR).**    Given a dataset $\mathcal{D} = \{(\mathbf{x}_i, y_i)\}_{i=1}^{N}$ with $\mathbf{x}_i \in \mathbb{R}^d$ and $y_i \in \mathbb{R}$, SR seeks a function $f : \mathbb{R}^d \to \mathbb{R}$ that minimizes a loss function $\mathcal{L}$, which is typically the empirical mean squared error (MSE). The discovered $f$ should be concise and human-interpretable. Some works additionally penalize expression complexity when selecting the final function (Cranmer, 2023).

**LLM-guided equation proposal.**    Let $\mathcal{P}$ be the space of prompts that encode the task description and a few-shot set of equation exemplars. These exemplars are concrete equation candidates previously discovered during the search and stored in the exemplar buffer (see Section 4.4 for details). Let $\mathcal{T}$ be the space of candidate equation *templates*, and an equation template $\tau \in \mathcal{T}$ is an equation form whose symbolic operations are fixed while the numeric constants are left unspecified. Later, we fit these constants to data, turning the template into an executable equation. In practice, we represent each template as a short Python code snippet. Conditioned on $p \in \mathcal{P}$, an LLM induces a distribution $q_p(\cdot)$ over $\mathcal{T}$. Given $k$ i.i.d. samples $\tau_i \sim q_p$, $i = 1, \ldots, k$, we define the expected best-of-$k$ loss as

$$\Phi_k(p) = \mathbb{E}_{\tau_i \sim q_p,\ i=1,\ldots,k} \left[ \min_{1 \le j \le k} \mathcal{L}(\tau_j) \right], \tag{1}$$

where $\mathcal{L}(\tau_j)$ is the fitted loss for template $\tau_j$ after parameter optimization, and best-of-$k$ denotes selecting the loss of the best candidate among $k$ independent templates sampled from $q_p$. Conceptually, one may seek a prompt $p^\star$, such that

$$p^\star \in \arg\min_{p \in \mathcal{P}} \Phi_k(p). \tag{2}$$

To this end, finding a better prompt is a way to optimize the final result. However, a static few-shot prompt biases the LLM's search distribution, $q_p$, towards regions merely syntactically similar to the exemplars, which may not correspond to regions of low empirical loss. Instead, we construct prompts adaptively based on the evolving exemplar buffer, which can be seen as a form of meta-optimization over $p$ that gradually shifts $q_p$ toward the template with lower empirical loss.

## 3    RELATED WORK

**Search-based methods**    explore equation space via stochastic operators. Genetic programming (GP) (Koza & Poli, 2005; Dubčáková, 2011; Cranmer, 2023) evolves expression trees through

crossover and mutation, while RL-based methods cast SR as a sequential decision process (Petersen et al., 2021; Xu et al., 2024). However, both suffer from the combinatorial explosion of the search space—leading to slow convergence, hyperparameter sensitivity, and overly complex expressions (La Cava et al., 2021; Holt et al., 2023). Critically, they require evaluating hundreds of thousands of candidates, making them computationally expensive.

**LLM-based search and refinement methods**   utilize LLMs to encode strong priors and symbolic reasoning into search-based SR (Shojaee et al., 2025a; Wang et al., 2025), enabling more efficient exploration of the hypothesis space. In these approaches, candidate equations are represented as executable code, evaluated on data, and then fed back to the LLM, which is prompted to produce improved versions. Some methods further augment LLM prompting with error-related signals (e.g., loss values or residuals) to provide a weak signal (Ma et al., 2024; Wang et al., 2025). More recent frameworks, such as (Grayeli et al., 2024; Wang et al., 2025), also maintain a dynamic concept library that summarizes validated patterns and leverages them to bias future generations toward more promising regions of the search space. Unlike prior work that samples many positive and negative hypotheses and abstracts them into natural-language concepts, RESTART distills structure only from high-value generations and stores it as executable code, yielding more efficient and directly actionable guidance.

**Mapping-based methods**   treat SR as supervised sequence prediction, training autoregressive models on large synthetic corpora (Petersen et al., 2021; Biggio et al., 2021; Kamienny et al., 2022; Li et al., 2023). They deliver strong single-pass hypotheses but are brittle under distribution shift. Several works finetune mapping-based SR models with RL objectives (Landajuela et al., 2022; Kamienny et al., 2023) to improve generalization via error-driven updates. However, this approach is highly sample-inefficient and gradient updates on large language models are computationally prohibitive, making it impractical at LLM scale. This motivates coupling mapping-based initialization with guided, iterative refinement—precisely what our framework achieves.

## 4    METHODOLOGY

Our proposed **RESTART** implements the complete *hypothesize–experiment–identify–improve* cycle by unifying a powerful LLM-based generator with a principled refinement mechanism, as shown in Figure 1. Its three core components are:

- *Informative Initialization*: The search begins with a strong hypothesis from a mapping-based estimator, preserving nonlinear structures that linear models often miss.
- *Targeted Refinement*: The unexplained signal from each experiment is explicitly modeled as an error-aware subproblem. The solution to this subproblem directly guides the LLM's next hypothesis.
- *Cumulative Knowledge Retention*: Successful, boosting-driven revisions are distilled into a reusable structure library, enabling long-term knowledge accumulation across iterations.

Unlike prior work that uses static few-shot prompts, RESTART adaptively constructs prompts by incorporating both short-term feedback and long-term knowledge.

### 4.1    INITIALIZATION VIA TRANSFORMER

Rather than initializing the search with a simple linear model (Shojaee et al., 2025a), which only captures additive effects of the input variables, we initialize with the transformer-based estimator E2E (Kamienny et al., 2022). E2E directly maps the input data to a symbolic expression as a prior. This data-dependent initialization yields a stronger initial hypothesis $f_0$ that already encodes salient nonlinearities, such as polynomial, trigonometric, and exponential functions, as well as interaction terms between multiple variables.

### 4.2    INSPECTING SYMBOLIC SUBPROBLEMS

At iteration $t = 0, 1, \ldots$, let $f_t : \mathbb{R}^d \to \mathbb{R}$ denote the current symbolic hypothesis. We consider a symbolic function class $\mathcal{G}$ consisting of exploration functions $g : \mathbb{R}^{d+1} \to \mathbb{R}$ that take as input both

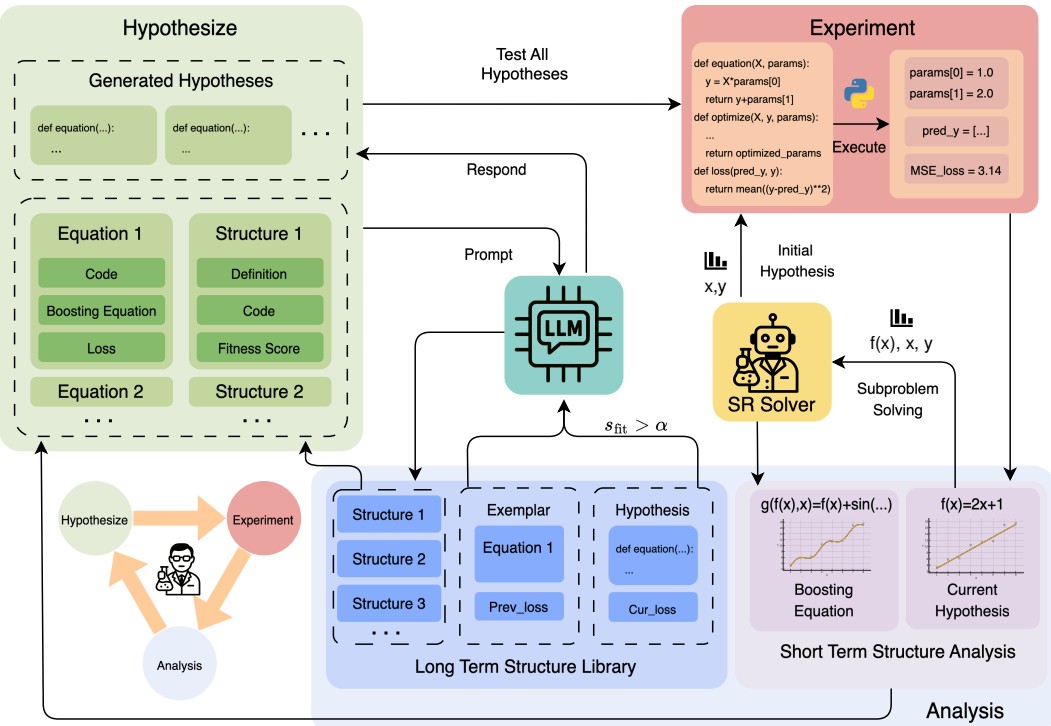

Figure 1: **Overview of the RESTART framework.** The framework follows three main steps of the scientific cycle (illustrated at bottom left): (i) **Hypothesis stage:** We first generate an informative initial hypothesis using a symbolic regression solver. In subsequent iterations, RESTART adaptively constructs prompts from exemplars and the structure library, and queries the LLM to generate several hypotheses $f(x)$ as executable Python functions. (ii) **Experiment stage:** RESTART executes each $f(x)$ on the dataset $(X, y)$, computes the loss, and optimizes its parameters (constants) to minimize the error. (iii) **Analysis stage:** RESTART formulates a subproblem $g(f(x), x)$ as a targeted refinement, solves it using a symbolic regression solver, and adds both $f(x)$ and $g(f(x), x)$ to the exemplar buffer. It then computes a fitness score $s_{\text{fit}}$; if $s_{\text{fit}} > \alpha$, we use the LLM to summarize the structure that led to the improvement and store it in the structure library for knowledge accumulation.

the prediction $f_t(\mathbf{x})$ at iteration $t$ and the original features $\mathbf{x}$:

$$g_t \ \leftarrow \ \arg\min_{g \in \mathcal{G}} \ \mathcal{L}\big(g(f_t(\mathbf{x}), \mathbf{x}), y\big), \tag{3}$$

where $\mathcal{L}(g(\cdot), y)$ denotes the loss between predictions $g(\cdot)$ and targets $y$. The objective in Eq. 3 makes the exploration function $g_t$ a boosting-style correction of the hypothesis $f_t$ toward the target $y$. Because $g_t$ takes both $f_t(\mathbf{x})$ and $\mathbf{x}$ as inputs, the symbolic form of $g_t(f_t(\mathbf{x}), \mathbf{x})$ describes how the current prediction should be adjusted as a function of its current value and the features in order to better match the data.

The backend for solving Eq. 3 is pluggable; viable options include KAN (Liu et al., 2025), RL-based methods (Petersen et al., 2021), or mapping-based methods (Kamienny et al., 2022). The choice of backend does not alter the formulation (see Appendix B.3 for details). In our experiments, we adopt a pre-trained Transformer (Kamienny et al., 2022) to balance speed and accuracy. Finally, we store the tuple $S_t = (f_t, \mathcal{L}(f_t(\mathbf{x}), y), g_t, \mathcal{L}(g_t(f_t(\mathbf{x}), \mathbf{x}), y))$ in the exemplar buffer $\mathcal{B}_t$ to guide subsequent prompt construction.

### 4.3 RETAINING VALIDATED IMPROVEMENTS

While structure analysis identifies a hypothesis's shortcomings, applying these insights indiscriminately risks overfitting to short-term noise, inflating complexity, and relying too heavily on approximate equations. To mitigate this, we retain only those boosting-driven updates that demonstrably

reduce the task loss. This process involves two steps: (i) gating modifications with a fitness score, and (ii) distilling eligible modifications into reusable structures.

**Step 1: Evidence and gating.** To ensure only meaningful modifications are retained, we gate updates using a comprehensive fitness score, $s_{\text{fit}}$. This score holistically evaluates a candidate equation's improvement by considering both relative and absolute loss reduction compared to exemplars. Given a new hypothesis $f_{\text{new}}(x)$ with loss $l_{\text{new}} = \mathcal{L}(f_{\text{new}}(x))$ and the exemplar hypothesis $f_{\text{base}}(x)$ with the lowest loss $l_{\text{base}} = \mathcal{L}(f_{\text{base}}(x))$ from the current prompt, we compute,

$$R = \frac{l_{\text{base}} - l_{\text{new}}}{l_{\text{base}}}, \text{ and } \Delta = l_{\text{base}} - l_{\text{new}},$$

where $R$ is the relative improvement ratio and $\Delta$ is the absolute improvement. To compare values across different scales, we normalize them to a range of $[0, 1)$ as,

$$s_r = 1 - e^{-kR}, \text{ and } s_a = 1 - (1 + \Delta)^{-1}.$$

Here, $s_r$ captures the proportional gain, with $k$ as a hyperparameter that rescales its sensitivity, while $s_a$ captures the absolute gain in a saturating form, preventing extremely large absolute improvements from dominating the score. Then we combine these scores using a weighted average, $s_{\text{fit}} = 100 \cdot (w_r s_r + w_a s_a)$, where $w_r + w_a = 1$ (see Appendix B.1 for more details). This design recognizes both large absolute gains on high-error problems and significant relative gains on low-error ones.

**Step 2: Structure formation.** After scoring each candidate modification $f_t \rightarrow f_{t+1}$ with $s_{\text{fit}}$, given a threshold $\alpha$, we retain those with $s_{\text{fit}} \geq \alpha$ and refer to them as *high-value*. For any high-value modification, the associated structural change often captures information about the ground-truth equation, which may help explain the high fitness score and can benefit subsequent iterations. Accordingly, we prompt the LLM with $\left(f_t, f_{t+1}, \mathcal{L}(f_t(\mathbf{x}), y), \mathcal{L}(f_{t+1}(\mathbf{x}), y)\right)$ to summarize the salient change as

$$c_{t+1} = (\texttt{name}, \texttt{desc}, h),$$

where `name` is a canonical identifier, `desc` is a brief textual description, and $h$ is a small Python code snippet (e.g., `np.sin(x)`) implementing the structure. Representing each structure as a code snippet aligns with our design: the structure is a distilled summary of the observed modification and can directly guide subsequent generations. We update the structure library $C$ by inserting $(c_{t+1}, s_{\text{fit}})$ and, whenever another entry shares the same `name`, merging them by (i) taking the set union of their code snippets $h$, (ii) assigning to the merged entry the highest $s_{\text{fit}}$ observed across its instances, and (iii) keeping the earliest `desc` among those instances. We enforce a capacity constraint $|C| \leq K$ and, if exceeded, evict the lowest-scoring entries until the constraint is satisfied, thereby limiting redundancy while keeping the library compact and effective.

## 4.4 Adaptive prompt construction

At each iteration, the prompt for generating new hypotheses $f_{t+1,j}$ integrates three complementary information sources: (i) few-shot exemplars, (ii) boosting equation summaries, and (iii) validated candidate structures. This design couples short-term feedback with long-term, reusable knowledge.

**Few-shot exemplars.** We maintain an exemplar buffer $\mathcal{B}_t$ as populations with multiple islands, suggested by Shojaee et al. (2025a) and Grayeli et al. (2024) (see Appendix B.1 for more details). For iteration $t+1$, we select $n$ exemplars $\{S_{t+1}^{(i)}\}_{i=1}^n \subset \mathcal{B}_t$ via MSE-weighted random sampling to balance quality and diversity. Each exemplar includes the fitted equation and a compact summary of its associated boosting equation, which highlights the residual error. This provides immediate context on what has been learned and what remains unexplained.

**Structure snippets.** To inject higher-level, reusable guidance, we sample $m$ structures from the library $\mathcal{C}_t$ with a probability proportional to their validation score. Each structure provides a descriptive name, a short definition, and a symbolic code sketch. While boosting equations offer instance-specific, short-term cues, the structure library encodes cumulative, and long-term refinements.

**Adaptive Hypothesis Generation Prompt**

**1. Task Definition:**
The prompt begins by setting the LLM's role, overall goal, and the expected output format.

**2. Few-Shot Exemplar:**
Provides a concrete example of a past attempt, including the crucial boosting equation analysis in its docstring.

**3. Iterative Context & Final Instruction:**
The final section provides the most recent context, including high-scoring structures, and gives the final instruction for generating the next equation.

```
# You are a helpful assistant ...
# Complete the 'equation' function below, ...
# Find the mathematical function skeleton that
    represents ...

def equation_v0(X, params):
 '''
 Analysis for current equation(f):
    current equation f(X) with MSE: ...
    the boosting equation g(f(X),X) found to be:
        g_0(X) = ...
 with MSE: ...
 '''

 ...
 return ...

# ... (Rest of the exemplars) ...

def equation_vj(X, params):
    '''
    Improved version of 'equation_vj-1'.
    Top scoring structures for the problem:
        - Name1, fit: ...
            definition: ...
            functions: ...
        ...
    Consider previous versions of the equation with
     the corresponding boosting equation function
     and structures(if any) for the problem ...
    '''
```

Figure 2: The annotated structure of our adaptive prompt. The prompt is dynamically assembled in three main parts: (1) A fixed **task definition** setting the goal. (2) A series of **few-shot exemplars** showing past attempts and their boosting equation analyses. (3) The final **iterative context**, which includes a reusable structure library and the instruction for generating the next hypothesis.

**Prompt assembly.** The final prompt is constructed from a fixed template populated with: (i) the task definition and variable specifications, (ii) the $n$ exemplar–boosting equation pairs, and (iii) the $m$ high-scoring structure snippets (see Figure 2). Conditioned on this prompt $p_{t+1}$, the generator LLM $q$ autoregressively samples $k$ candidate symbolic expressions. Each candidate is emitted as a Python function with tunable constants (e.g., def f(x, params)). We then optimize these constants using a BFGS solver to minimize the training loss, yielding a set of fitted hypotheses $\{f_{t+1,j}\}$. Each new hypothesis, along with its loss and boosting analysis, is appended to the buffer $\mathcal{B}_{t+1}$, ensuring subsequent prompts reflect the evolving understanding. This approach tightly integrates short-term error signals with long-term structural motifs. Though the per-hypothesis analysis adds computational overhead, it provides precise feedback that significantly improves sampling efficiency. Our experiments show that this leads to faster convergence and superior solutions without a substantial increase in total runtime when using the E2E (Kamienny et al., 2022) backend.

## 5 EXPERIMENTS

We present a comprehensive evaluation of our proposed framework, RESTART, designed to validate the effectiveness of its end-to-end hypothesize–experiment–analyze loop. Our experiments demonstrate that its principled feedback mechanisms achieve consistently improved performance.

| | LSR-Transform | | Biology | | Material Science | | Physics | |
|---|---|---|---|---|---|---|---|---|
| | NMSE | ACC | NMSE | ACC | NMSE | ACC | NMSE | ACC |
| DSR | $0.472_{\pm 1.755}$ | $36.04_{\pm 48.23}$ | $0.206_{\pm 0.278}$ | $16.67_{\pm 38.07}$ | $0.058_{\pm 0.099}$ | $36.00_{\pm 48.99}$ | $0.104_{\pm 0.111}$ | $27.27_{\pm 45.05}$ |
| E2E | $2.098_{\pm 10.794}$ | $57.66_{\pm 49.63}$ | $0.578_{\pm 0.400}$ | $12.50_{\pm 33.78}$ | $0.008_{\pm 0.016}$ | $76.00_{\pm 38.51}$ | $0.425_{\pm 0.430}$ | $20.45_{\pm 34.64}$ |
| PySR | $0.175_{\pm 0.390}$ | $72.07_{\pm 44.56}$ | $0.003_{\pm 0.012}$ | $54.17_{\pm 50.90}$ | $0.057_{\pm 0.282}$ | $72.00_{\pm 45.83}$ | $0.004_{\pm 0.010}$ | $73.86_{\pm 36.55}$ |
| LLMDirect | $0.355_{\pm 0.376}$ | $32.88_{\pm 46.95}$ | $0.454_{\pm 0.379}$ | $16.67_{\pm 38.07}$ | $0.012_{\pm 0.018}$ | $58.00_{\pm 47.17}$ | $0.099_{\pm 0.245}$ | $36.36_{\pm 48.66}$ |
| LLMSR | $0.160_{\pm 0.353}$ | $74.32_{\pm 42.57}$ | $0.016_{\pm 0.053}$ | $70.83_{\pm 44.03}$ | $0.003_{\pm 0.015}$ | $\mathbf{96.00_{\pm 20.00}}$ | $\mathbf{0.002_{\pm 0.008}}$ | $84.09_{\pm 37.00}$ |
| SGA | $0.374_{\pm 0.579}$ | $37.39_{\pm 45.46}$ | $0.975_{\pm 2.587}$ | $12.50_{\pm 33.78}$ | $4.021_{\pm 19.996}$ | $48.00_{\pm 48.99}$ | $0.345_{\pm 1.260}$ | $34.09_{\pm 45.46}$ |
| RESTART | $\mathbf{0.157_{\pm 0.407}}$ | $\mathbf{74.77_{\pm 42.04}}$ | $\mathbf{0.001_{\pm 0.005}}$ | $\mathbf{77.08_{\pm 38.95}}$ | $\mathbf{0.001_{\pm 0.002}}$ | $\mathbf{96.00_{\pm 20.00}}$ | $0.003_{\pm 0.009}$ | $\mathbf{85.23_{\pm 35.07}}$ |

Table 1: Performance comparison on 4 datasets within training distribution. RESTART achieves the best or near-best performance across all datasets. For LLM-based approaches, the backbone model is Qwen3-8B (Yang et al., 2025).

## 5.1 EXPERIMENTAL SETUP

LLM-SRBench (Shojaee et al., 2025b) is a comprehensive benchmark designed to evaluate LLM-based scientific equation discovery methods beyond simple memorization. We focus on two key categories from this benchmark: i) LSR-Transform, which contains 111 problems derived from established physical models; ii) LSR-Synth, which includes 93 problems that combine known scientific terms with novel, plausible synthetic terms from biology, physics, and materials science domains (see details in Appendix E).

To contextualize the performance of RESTART, we compare it against representative baselines from major symbolic regression paradigms: PySR (Cranmer, 2023) (GP-based), DSR (Petersen et al., 2021) (RL-based), E2E (Kamienny et al., 2022) (mapping-based), LLMDirect (Shojaee et al., 2025b) (LLM baseline), SGA Ma et al. (2024) (LLM optimization), and LLMSR (Shojaee et al., 2025a) (Current SOTA LLM-based SR). For LLM-based methods, we use Qwen3-8B (Yang et al., 2025) as the default LLM backbone, unless stated otherwise. Hyperparameters follow the authors' public settings where available (see Appendix D for details).

### 5.1.1 EVALUATION METRICS

Following (Shojaee et al., 2025a;b), we evaluate numeric fit quality using the normalized mean squared error (NMSE): $\text{NMSE} = \frac{\sum_{i=1}^{N_{\text{test}}}(\hat{y}_i - y_i)^2}{\sum_{i=1}^{N_{\text{test}}}(y_i - \bar{y})^2}$, where $y_i$ denotes the true target for test input $x_i$, $\hat{y}_i$ the corresponding model prediction, $\bar{y} = \frac{1}{N_{\text{test}}}\sum_{i=1}^{N_{\text{test}}} y_i$ the mean of the test targets, and $N_{\text{test}}$ the number of test samples. To ensure numerical stability—particularly when exponential or logarithmic terms cause extreme values or the denominator approaches zero—we clip the NMSE at a maximum of 100. This clipping prevents exploding errors and division-by-near-zero effects from skewing the averages, enabling more robust cross-task comparisons.

Following common practice in SR benchmarks (Shojaee et al., 2025a;b), we report the Accuracy to tolerance $\tau$ ($Acc_\tau$), which measures whether the discovered equation matches the ground-truth expression across the entire input domain: $Acc_\tau = \mathbb{1}\left(\max_{1 \leq i \leq N_{\text{test}}} \left| \frac{\hat{y}_i - y_i}{y_i} \right| \leq \tau \right)$, with $\tau = 0.1$ unless stated otherwise. This metric evaluates generalization by requiring uniformly low relative error across all data points, rather than just low average error, which can mask overfitting. To ensure robust performance estimates and mitigate the influence of pathological outliers, we follow (Kamienny et al., 2022; Biggio et al., 2021) by reporting results over the best 95% of predictions, discarding the worst 5% based on relative error.

## 5.2 MAIN EXPERIMENTAL RESULTS

**Research Question 1: Does RESTART discover accurate symbolic expressions from data?**
Quantitative results in Table 1 show that RESTART achieves superior performance, with lower NMSE and higher accuracy than all baseline methods across nearly every benchmark. This establishes its effectiveness in discovering accurate symbolic expressions. A focused analysis on the three most challenging tasks from the Biology dataset shown in Figure 3 reveals that RESTART's relative advantage is greatest under difficult conditions, particularly for problems BPG12 and BPG20. The minimal NMSE values suggest a successful identification of the underlying equation's structure.

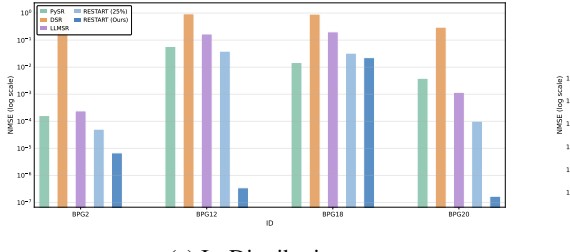 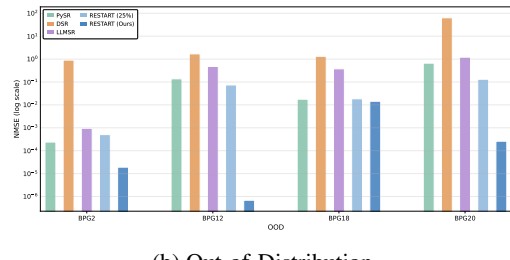

(a) In-Distribution       (b) Out-of-Distribution

Figure 3: NMSE of RESTART on four challenging problems under (a) in-distribution and (b) out-of-distribution settings. In both settings, RESTART not only performs well on problems solvable by baselines but also yields notable improvements on these more complex tasks.

| | Biology | | Material Science | | Physics | |
|---|---|---|---|---|---|---|
| | NMSE | ACC | NMSE | ACC | NMSE | ACC |
| DSR | $29.900_{\pm43.344}$ | $16.67_{\pm38.07}$ | $7.348_{\pm20.664}$ | $84.00_{\pm37.42}$ | $5.715_{\pm21.184}$ | $25.00_{\pm43.80}$ |
| E2E | $39.465_{\pm45.092}$ | $4.17_{\pm20.41}$ | $1.395_{\pm2.172}$ | $94.00_{\pm21.98}$ | $25.310_{\pm39.610}$ | $21.59_{\pm36.40}$ |
| PySR | $8.875_{\pm26.612}$ | $37.50_{\pm49.45}$ | $4.115_{\pm19.980}$ | $96.00_{\pm20.00}$ | $\mathbf{7.151}_{\pm23.052}$ | $62.50_{\pm43.30}$ |
| LLMDirect | $68.661_{\pm44.966}$ | $12.50_{\pm33.78}$ | $1.704_{\pm3.801}$ | $84.00_{\pm34.52}$ | $28.579_{\pm44.478}$ | $13.64_{\pm34.71}$ |
| LLMSR | $6.667_{\pm19.165}$ | $45.83_{\pm50.90}$ | $0.084_{\pm0.239}$ | $96.00_{\pm20.00}$ | $14.808_{\pm34.460}$ | $65.91_{\pm47.95}$ |
| SGA | $63.701_{\pm45.904}$ | $8.33_{\pm28.23}$ | $12.072_{\pm29.957}$ | $78.00_{\pm38.41}$ | $33.410_{\pm44.259}$ | $14.77_{\pm35.07}$ |
| RESTART | $\mathbf{5.087}_{\pm14.570}$ | $\mathbf{52.08}_{\pm47.73}$ | $\mathbf{0.075}_{\pm0.198}$ | $\mathbf{100.00}_{\pm0.00}$ | $8.167_{\pm26.233}$ | $\mathbf{71.59}_{\pm44.98}$ |

Table 2: Performance comparison on 3 datasets (OOD results only). NMSE values are clipped at a maximum of 100. For LLM-based approaches, the backbone model is Qwen3-8B (Yang et al., 2025).

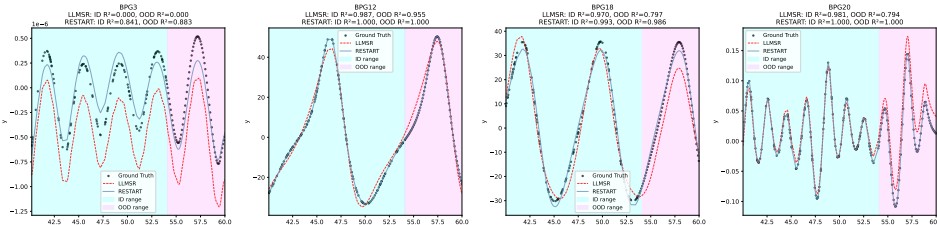

Figure 4: **Visualizing equations on challenging tasks.** Each panel shows the ground-truth equation (black) and the outputs from LLMSR (red) and **RESTART** (blue). We select four challenging equations from the Biology dataset and evaluate the discovered equations over both in-distribution (blue shaded) and out-of-distribution (pink shaded) regions. RESTART not only matches the ground truth within the training range but also generalizes well to unseen regions, closely following the true functional shape.

Qualitative analysis confirms this: baseline methods frequently omit key terms, whereas RESTART's adaptive refinement iteratively incorporates them. This highlights the core benefit of our short-term guidance, where the exploration function $g_t$ provides data-driven signals for precise corrections. For example, RESTART correctly identified the harmonic interaction in BPG12 as shown in Figure 4, a component missed by LLMSR. (An analysis of the refinement process is provided in Appendix H)

**Research Question 2: Does RESTART recover equations close to the ground truth?** On out-of-distribution (OOD) splits, RESTART achieves the highest accuracy across all domains as shown in Table 2. This demonstrates that our method not only fits the training data well but also discovers equations that generalize effectively to unseen data.

Figure 4 visualizes the equations discovered on a subset of challenging tasks. For problems like BPG18 and BPG20, baselines such as LLMSR may achieve low training error, but their solutions visibly diverge outside the training region. This divergence is often caused by overly complex terms that reduce local error but distort the global function shape. In contrast, RESTART 's short-term guidance mechanism explicitly identifies missing operators or interactions, which prevents the model from fitting noise and yields more robust equations. Furthermore, the improvement-gated

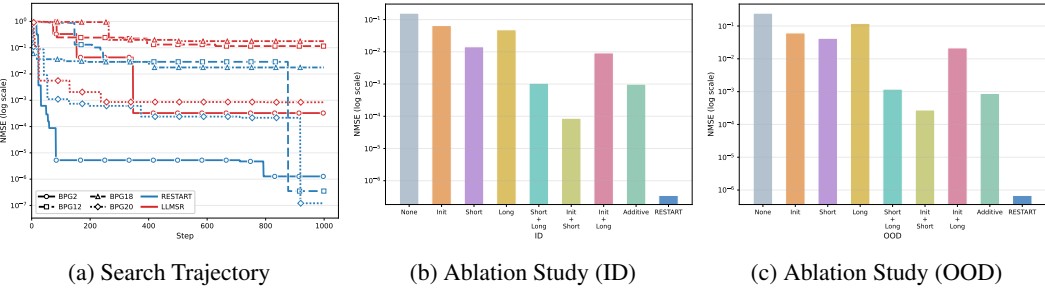

(a) Search Trajectory          (b) Ablation Study (ID)          (c) Ablation Study (OOD)

Figure 5: (a) The median search trajectories on four challenging equations; (b) Ablation study on BPG12 under ID setting; (c) Ablation study on BPG12 under OOD setting.

structure library retains only validated components, preventing equation bloat and ensuring the final expressions are both concise and physically meaningful.

**Research Question 3: Is RESTART computationally efficient?**   Although our guidance mechanisms introduce overhead for subproblem solving and structure summarization, RESTART is significantly more efficient at discovering accurate equations. As shown in Figure 5a, RESTART's NMSE decreases sharply within the first 100 iterations, rapidly surpassing the performance of LLMSR. This swift convergence demonstrates that the cost of targeted analysis is substantially offset by a major gain in search efficiency. By guiding the search toward promising regions instead of exploring blindly, RESTART drastically reduces the number of ineffective hypotheses evaluated.

Furthermore, Figure 3 shows that RESTART discovers better equations than other baselines using only 25% of the iterations. This underscores its ability to find strong, generalizable solutions with a fraction of the computational budget, proving that our approach achieves a superior trade-off between analysis overhead and overall search effectiveness.

### 5.2.1 ABLATION STUDY

**Research Question 4: What is the individual contribution of each key component: initialization, short-term guidance, and long-term memory?**   We conduct an ablation study to evaluate the contributions of RESTART along three axes: (i) removing key components from RESTART, (ii) evaluating different LLM backbones, and (iii) using alternative subproblem solvers. The results, shown in Figure 5b and 5c, confirm that all three components are critical for peak performance. Removing any one—initialization, short-term analysis, or long-term retention—causes a noticeable performance drop, validating the necessity of the full hypothesize–experiment–analyze loop. Additionally, we evaluated an alternative variant of RESTART, termed *Additive*, whose purpose is to test the behavior of the framework when the boosting step is restricted to fitting the classical additive residual. In this variant, the exploration function is defined as $g : \mathbb{R}^n \rightarrow \mathbb{R}$ and is fit as $g(x) = y - f_t(x)$. After solving this subproblem, we prompt the LLM with $\left( f_t,\ \mathcal{L}(f_t(\mathbf{x}), y),\ g_t,\ \mathcal{L}(g_t(\mathbf{x}),\ y - f_t(\mathbf{x})) \right)$, so that the model receives both the current hypothesis and the residual-based signal for comparison with the full RESTART formulation. From Figures 5b and 5c, we observe a substantial increase in NMSE under the Additive variant. This highlights that performance improvements in RESTART arise primarily from structural refinement rather than simply filling the numerical residual gap.

Furthermore, while the choice of LLM backbones affects absolute performance (with stronger models yielding higher accuracy, as seen in Table 3), RESTART consistently outperforms other LLM-based methods regardless of the foundation model. This indicates that its advantage stems from the algorithmic design rather than a specific model's capabilities.

Finally, as shown in Table 4, we observe a trade-off in subproblem solvers: more expressive solvers can propose better structures at a higher computational cost. However, results in Tables 1 and 2 show that a traditional solver alone is insufficient. Optimal symbolic recovery—achieving both accuracy and parsimony—requires its integration with RESTART 's iterative hypothesis refinement and guidance mechanisms. Details are provided in the Appendix B.3

### 5.2.2 Case Study

**Research Question 5: How does RESTART perform on a real-world scientific task?**   Here, we consider a real-world scientific discovery task. Defining and measuring particle speed in the classically forbidden region is crucial for testing Bohmian mechanics and understanding microscopic transport. Using coupled waveguides, Sharoglazova et al. (2025) showed that the measured nondirectional speed follows $v = \sqrt{2|\Delta|/m}$, where $v$ is the particle speed, $m$ the particle mass, and $\Delta$ the energy offset inside the step ($\Delta = E - V_0 + \hbar J_0$). This relation describes how speed increases with $|\Delta|$ in the forbidden region and is mirror-symmetric in $\Delta$. While the equation appears elementary, the actual physical law used in the experiment is: $v = \sqrt{\frac{2|\Delta \cdot 1.602176634 \times 10^{-22}|}{m}} \Big/ 1000$, which is more challenging. It involves (i) nested nonlinear operators (absolute value within a square root), (ii) cross-variable interaction through division, and (iii) large-magnitude unit-conversion constants embedded inside nonlinear transformations, which makes constant recovery and operator ordering difficult for search-based SR.

To assess RESTART in a real-world scenario, we re-examine this physical law from only 46 experimental data points, including basic unit conversions (e.g., meV→J). As shown in Figure 6a and Figure 6b, RESTART successfully rediscovers the relationship and produces two key expressions: **RESTART-1** achieves the best fit by recovering the analytic scaling while adding compact, interpretable corrections (e.g., parameterized scaling and energy-dependent terms) that may account for known systematics in the speed extraction process (the parabolic build-up fit can overestimate $v$ by up to $\sim 6.7\%$ in the forbidden regime (Sharoglazova et al., 2025)); **RESTART-2** more closely adheres to the published formula, sacrificing a small amount of fit quality for greater generalizability and interpretability. These results demonstrate that RESTART is highly data-efficient and prior-aware. Through short-term refinement, it can generate targeted, testable corrections that help bridge theory and experiment, even with scarce, noisy data. To further validate RESTART 's capability in scientific discovery, we report a more challenging task in Appendix G.

| | $R^2$(data) | $R^2$(equation) |
|---|---|---|
| $v = \sqrt{\dfrac{2|\Delta|}{m}}$ | 0.9642 | 1.000 |
| RESTART-1 | 0.9827 | 0.9699 |
| RESTART-2 | 0.9672 | 0.9985 |
| LLMSR | 0.9688 | 0.9958 |
| LLMDIRECT | 0.0 | 0.0 |
| PYSR | 0.8984 | 0.0 |

(a) Energy results table

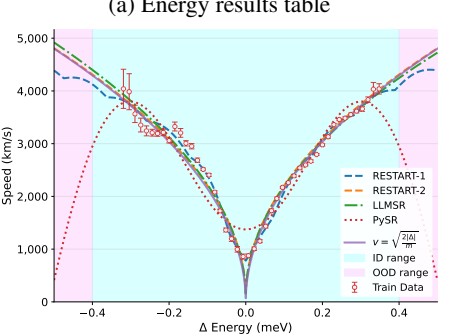

(b) Energy figure

## 6 Discussion and Conclusion

We presented RESTART, a symbolic regression framework that completes the hypothesize–experiment–analyze cycle by combining LLM-guided hypothesis generation with structural analysis and a persistent structure library. Our experiments demonstrate that the explicit analysis stage is critical: solving the subproblem $g(f(x), x)$ and using it to guide the LLM with structural refinements, together with the structure library, leads to a more efficient and accurate search process. The physics case study further illustrates the potential of symbolic regression for real scientific tasks.

Figure 6: (a) Comparison of two RESTART-discovered equations with the reported equation $v = \sqrt{\frac{2|\Delta|}{m}}$ (Sharoglazova et al., 2025) and baselines. $R^2$(data) denotes the $R^2$ computed against the measured ground-truth data points, whereas $R^2$(equation) denotes the $R^2$ from points sampled via the reported equation. Both discovered equations achieve a higher $R^2$ on experimental data while remaining consistent with the reported equation, suggesting a plausible correction. (b) Visualization of three equations: RESTART-1 fits the data well but slightly diverges from the reported equation in the OOD range.

Remaining challenges include the computational cost of repeated LLM queries and the inability to directly verify the correctness of each stored structure. Future work will explore integrating auxiliary models to pre-filter structures and candidate hypotheses, as well as incorporating domain constraints to improve plausibility without increasing computational overhead.

ACKNOWLEDGMENTS

The research work described in this paper was conducted in the JC STEM Lab of Machine Learning and Symbolic Reasoning funded by The Hong Kong Jockey Club Charities Trust.

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

## A  LLM USAGE

We used LLMs to assist in the preparation of this manuscript. Specifically, LLMs were used for polishing the language, improving readability, and restructuring sentences for clarity. All conceptual contributions, experimental design, analyses, and results were produced and verified by the authors. The authors carefully reviewed and edited all model outputs before inclusion in the final manuscript.

## B  METHOD DETAILS

### B.1  IMPLEMENTATION AND EXPERIMENT DETAILS

We implement RESTART using a population-based exemplar buffer, following the island-based search paradigm (Shojaee et al., 2025a; Romera-Paredes et al., 2024; Cranmer, 2023). We maintain $M = 10$ disjoint islands, each storing a diverse population of candidate hypotheses $\{f_i(x)\}$ to promote exploration. At each iteration, we first select an island $\mathcal{I}$ uniformly at random, and then sample $k_{\text{exemplar}} = 4$ exemplars from this island to assemble the prompt.

Within each island, we group hypotheses according to their loss $L(f_i(x))$. Groups are sampled proportionally to a Boltzmann distribution (Maza & Tidor, 1993) over their losses:

$$p_i = \frac{\exp\big(L(f_i(x))/\beta_c\big)}{\sum_{j \in \mathcal{I}} \exp\big(L(f_j(x))/\beta_c\big)}, \qquad \beta_c = T_0 \left(1 - \frac{u \bmod N}{N}\right),$$

where $u$ denotes the number of hypotheses in island $\mathcal{I}$, $T_0 = 0.1$ is an initial temperature, and $N = 30000$ controls the annealing schedule. This scheme adaptively lowers the temperature as the island becomes saturated, gradually shifting from exploration to exploitation. Within each group, exemplars are sampled with a length-based probability, $p_i^{\text{len}} \propto \exp\left(-\frac{\text{len}(f_i)}{\beta_p}\right)$, where $\text{len}(f_i)$ is the program length and $\beta_p = 1$ is a hyperparameter that encourages shorter and more interpretable expressions. Newly generated hypotheses are inserted back into the same island for future iterations.

Additionally, experiments were conducted on two hardware setups: a workstation with $2\times$RTX 4090 GPUs and a server with $8\times$L40S GPUs.

**Improvement-Gated Scoring.**  To combine absolute and relative improvement signals, we define $s_{\text{fit}} = w_a \cdot \Delta + w_r \cdot R$ and $w_a + w_r = 1$, where $\Delta$ is the absolute improvement and $R$ is the relative improvement. The weights $w_a$ and $w_r$ are determined piecewise according to the logarithm of the absolute improvement $\Delta_{log} = \log_{10}(\Delta)$. We then set:

- If $\Delta_{log} \geq 1$: $w_a = 0.7$, $w_r = 0.3$.
- If $0 \leq \Delta_{log} < 1$: $w_a = 0.5$, $w_r = 0.5$.
- If $-2 \leq \Delta_{log} < 0$: $w_a = 0.2$, $w_r = 0.8$.
- If $\Delta_{log} < -2$: $w_a = 0.0$, $w_r = 1.0$.

This piecewise schedule ensures that large absolute improvements are weighted more heavily, whereas very small absolute improvements rely primarily on relative gains to be considered significant. After computing the fitted score $s_{\text{fit}}$, we regard an improvement as *valid* if $s_{\text{fit}} \geq \alpha$, $\alpha = 40.0$, and trigger the structure distillation process. To control memory and maintain efficiency, we cap the maximum number of structures stored in the library $K = 20$.

### B.2  LLM BACKBONE

We compare Qwen3 (Yang et al., 2025) backbones of varying scales and DeepSeek v3.1 (DeepSeek-AI et al., 2024) to assess how the generative prior affects symbolic search performance. Across most backbones, our method consistently outperforms LLM-SR, demonstrating the effectiveness of the structure analysis and targeted refinement. Stronger and reasoning-enabled models yield more accurate candidate programs, leading to improved equation discovery, but they also incur substantially higher computation cost. This comparison highlights a clear accuracy–compute trade-off: smaller backbones offer a good balance of speed and accuracy, whereas reasoning or larger backbones can further boost performance when additional compute budget is available.

| | Qwen3-1B | | Qwen3-8B | | Qwen3-8B-think | |
|---|---|---|---|---|---|---|
| | ID | OOD | ID | OOD | ID | OOD |
| LLMSR | 0.49983 | **0.69452** | 0.20248 | 0.37592 | 0.90352 | 1.22674 |
| RESTART | **0.45502** | 0.77328 | **0.02254** | **0.01442** | **0.00801** | **0.01103** |

| | Qwen3-32B | | Qwen-Flash | | DS-Chat | |
|---|---|---|---|---|---|---|
| | ID | OOD | ID | OOD | ID | OOD |
| LLMSR | $4.12 \times 10^{-4}$ | $3.09 \times 10^{-4}$ | 0.01499 | 0.02082 | 0.02884 | 0.04199 |
| RESTART | $1.34 \times 10^{-3}$ | $1.46 \times 10^{-3}$ | $1.76 \times 10^{-5}$ | $3.91 \times 10^{-6}$ | $1.70 \times 10^{-6}$ | $7.48 \times 10^{-7}$ |

Table 3: Comparison of LLMSR and RESTART across different LLM backbones on BPG18.

| | ID | OOD | Average Evaluation Time (s) |
|---|---|---|---|
| LLM | 0.02683 | 0.01481 | 6.418 |
| Polynomial | 0.47034 | 0.65663 | 0.580 |
| KAN | 0.03668 | 0.01963 | 27.651 |
| DSO | 0.01486 | 0.00849 | 59.91 |
| PySR | 0.00122 | 0.00096 | 14.153 |
| E2E | 0.02254 | 0.01442 | 6.587 |

Table 4: Comparison of different symbolic backbones on BPG18, along with the average time for the experiment and analysis stages per iteration (primarily subproblem solving). Note that the average time includes invalid evaluations, which contribute values close to zero.

## B.3 SUBPROBLEM SOLVER BACKEND

Table 4 provides a detailed comparison of candidate backends for solving the symbolic subproblem defined in Eq. (3). We evaluate representatives from mapping-based methods (E2E), search-based methods (PySR, Polynomial, DSR, KAN), a simple polynomial regression baseline, and a direct LLM-based solver that is prompted with the dataset $\{(f(x), X, y)\}$ to generate new equations. This LLM-based solver differs from the Llmdirect baseline in that it is data-aware rather than data-blinded. This comparison isolates the contribution of the subproblem solver from the rest of the RESTART pipeline and quantifies its impact on both accuracy and runtime.

Importantly, the choice of backend does not change the formulation of Eq. (3); it only affects the quality and efficiency of the discovered refinement $g_t$. This solver-agnostic design keeps the framework modular and allows it to benefit from future advances in symbolic regression without altering its overall search procedure.

Overall, we observe a clear **accuracy–compute trade-off**: more expressive search-based solvers (e.g., PySR, DSR) achieve lower NMSE but incur significantly higher computation cost, whereas fast mapping-based methods (E2E) sacrifice some accuracy in exchange for much higher throughput. Consistent with the results in Tables 1 and 2, a single solver alone is insufficient to recover optimal equations. The best performance is obtained when these solvers are embedded within the iterative hypothesize–experiment–analyze cycle of RESTART, where they provide targeted refinements rather than attempting to solve the entire problem from scratch.

## C RESULT DISTRIBUTION

Following the setup in LLM-SRBench (Shojaee et al., 2025b), we report box plots (Figure 7) that illustrate performance variations across the LLM-SRBench datasets for different methods, highlighting the distribution of the experimental results.

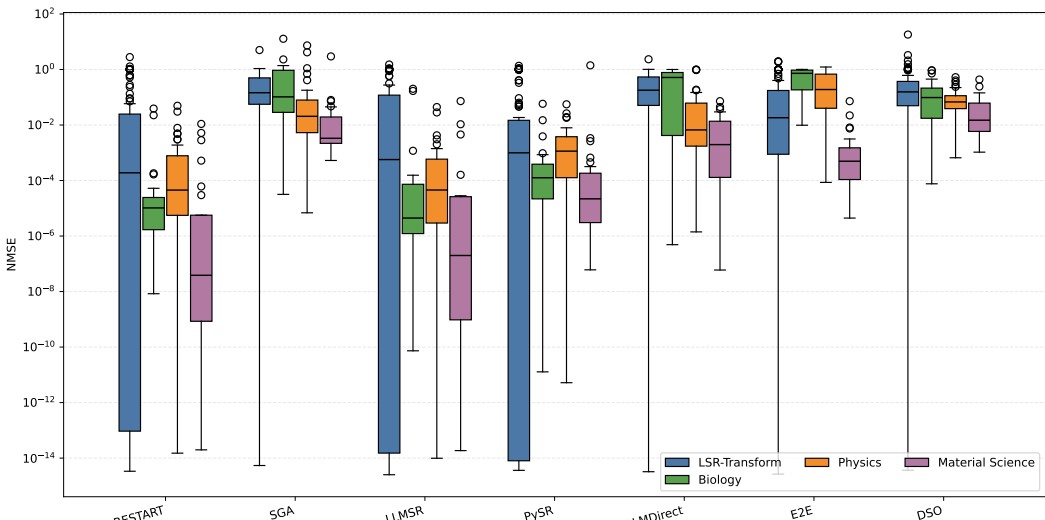

Figure 7: Distribution of in-distribution NMSE across all benchmark problems

# D    BASELINE METHOD DETAIL

## D.1    DSR

DSR[2] (Petersen et al., 2021) formulates symbolic expression generation as a sequential decision process in which a policy network constructs expressions token by token. DSR employs a risk-seeking policy-gradient objective that emphasizes learning from top-reward trajectories. Additionally, DSR trains a new model for each dataset. The hyperparameters are presented in Figure 8 and are used as the default settings unless otherwise specified.

## D.2    E2E

E2E[3] (Kamienny et al., 2022) employs a Transformer encoder–decoder that maps tokenized data directly to a symbolic expression string. The model is supervised on synthetic corpora to learn correspondences from data distributions to formula structures. During inference, autoregressive decoding with beam search outputs parseable expressions, followed by parameter fitting for numeric accuracy. The hyperparameters are presented in Figure 9 and are used as the default settings unless otherwise specified.

## D.3    PYSR

PySR[4] (Cranmer, 2023) uses a population-based genetic programming framework that represents candidate models as symbolic expression trees composed of primitive operators. It iteratively applies mutation, crossover, and island migration to evolve diverse candidates. A multi-objective criterion balances data-fit and expression complexity and maintains a Pareto front of solutions. The hyperparameters are presented in Figure 10 and are used as the default settings unless otherwise specified.

## D.4    LLMDIRECT

As suggested in Shojaee et al. (2025b), we include a data-blind zero-shot LLM baseline[5] that prompts to produce syntactically valid equations purely from its pretrained knowledge, without conditioning on the dataset. It relies on built-in mathematical priors and general reasoning to propose

---

[2]https://github.com/dso-org/deep-symbolic-optimization
[3]https://github.com/facebookresearch/symbolicregression
[4]https://github.com/MilesCranmer/PySR
[5]https://github.com/deep-symbolic-mathematics/llm-srbench

```
dsr_params:
 task:
   task_type: "regression"
   function_set: ["add", "sub", "mul", "div", "sin", "cos", "exp", "
       log"]
   metric: "inv_nrmse"
   metric_params: [1.0]
   extra_metric_test: null
   extra_metric_test_params: []
   threshold: 1.0e-12
   protected: false
   reward_noise: 0.0
   reward_noise_type: "r"
   normalize_variance: false
   decision_tree_threshold_set: []
 training:
   n_samples: 1000000
   batch_size: 10000
   epsilon: 0.05
   n_cores_batch: 50
 policy_optimizer:
   learning_rate: 0.0005
   entropy_weight: 0.03
   entropy_gamma: 0.7
 prior:
   length:
     min_: 4
     max_: 64
     "on": true
   repeat:
     tokens: "const"
     min_: null
     max_: 3
     "on": true
   inverse:
     "on": true
   trig:
     "on": true
   const:
     "on": true
   no_inputs:
     "on": true
   soft_length:
     loc: 10
     scale: 5
     "on": true
   domain_range:
     "on" : false
```

Figure 8: DSR hyperparameter configuration.

```
e2e_params:
 model_path: "e2e.pt"
 max_input_points: 200
 n_trees_to_refine: 10
 rescale: true
 max_num_samples: 2000
```

Figure 9: E2E hyperparameter configuration.

```
pysr_params:
 niterations: 40
 maxsize: 30
 populations: 15
 population_size: 33
 ncycles_per_iteration: 550
 binary_operators: ["+", "*", "-", "/", "^"]
 unary_operators: ["cos", "exp", "sin", "sqrt", "log"]
 batching: true
 batch_size: 5000
 constraints:
  "^": [-1, 20]
  "exp": 20
  "log": 20
  "sqrt": 20
  "sin": 10
  "cos": 10
```

Figure 10: PySR Hyperparameter Configuration

diverse symbolic forms. The hyperparameters are presented in Figure 11 and are used as the default settings unless otherwise specified.

```
llmdirect_params:
 samples_per_prompt: 5
```

Figure 11: LLMDirect hyperparameter configuration.

### D.5 SGA

SGA[6] (Ma et al., 2024) adopts a bilevel setup in which an LLM-based agent proposes symbolic structures while a separate optimizer fits the continuous parameters of each proposal. It also uses two different temperatures for LLM generation to separate exploitation and exploration. We follow the default hyperparameter settings suggested by Shojaee et al. (2025b) without additional modifications.

### D.6 LLMSR

LLMSR[7] (Shojaee et al., 2025a) is a state-of-the-art LLM-driven evolutionary method that generates equation candidates as code and refines them through LLM queries. It stores generated equations in populations; at each round, LLMSR randomly selects one population and samples equations to construct a new prompt. The hyperparameters are presented in Figure 12 and are used as the default settings unless otherwise specified.

```
llmsr_params:
 global_max_sample_num: 1000
 samples_per_prompt: 4
 num_islands: 10
```

Figure 12: LLMSR hyperparameter configuration.

---

[6] https://github.com/PingchuanMa/SGA
[7] https://github.com/deep-symbolic-mathematics/LLM-SR

## E    LLM-SRBENCH

LLM-SRBench (Shojaee et al., 2025b) is a benchmark designed for the systematic evaluation of symbolic regression (SR) methods, with a particular focus on leveraging large language models (LLMs) for scientific discovery. Unlike previous benchmarks such as SRBench (La Cava et al., 2021), where LLMs may simply recall the ground-truth equations from memory—thus bypassing the core purpose of the symbolic regression task—LLM-SRBench is constructed to be challenging, diverse, and representative of real-world scientific inference problems. Its design encourages LLMs to move beyond rote memorization and to demonstrate genuine reasoning and equation-discovery capabilities. The benchmark consists of two major categories:

1. **LSR-Transform:** This category contains 111 problems derived from established physical models such as kinematics, dynamics, and thermodynamics. Each problem is generated by systematically transforming known governing equations, for example by switching input and output variables, rearranging terms, or adding noise to simulate experimental conditions. This construction requires models to reason about the underlying functional relationships rather than merely recall canonical forms, thus providing a robust test of a model's capability for true equation discovery.

2. **LSR-Synth:** This category focuses on evaluating generalization. It includes problems that combine well-known scientific quantities with synthetic but physically plausible terms, ensuring that the solutions cannot be solved by rote memorization. The tasks span three scientific domains: biology, physics, and materials science. The presence of novel terms forces SR models to infer correct functional forms based on data patterns, simulating real-world scenarios where researchers encounter new variables or unmeasured effects.

Each problem in LLM-SRBench provides training and testing data sampled from the ground-truth function, as well as standardized evaluation metrics such as normalized mean squared error (NMSE). Together, these tasks cover both interpolation and extrapolation settings, making LLM-SRBench a comprehensive and challenging suite for benchmarking LLM-based SR algorithms.

## F    AVERAGE TRAJECTORY

As shown in Figure 13, we reported the average NMSE trajectories for four biology equations. Note that because NMSE values differ on a log scale, the mean can be dominated by the higher-NMSE trajectories. For example, if there are 10 trajectories—one with NMSE $1 \times 10^{-4}$ and nine with NMSE $1 \times 10^{-9}$—the average NMSE will still be above $1 \times 10^{-5}$, which is substantially higher than most of the trajectories.

## G    CASE STUDY

Here we report the discovered RESTART-1 and RESTART-2 in Figures 14 and 15.

| Method | $R^2$(equation) |
|---|---|
| RESTART | **0.9771** |
| PYSR | 0.5325 |
| LLMSR | 0.1737 |
| LLMDIRECT | 0.00 |

Table 5: Performance on the nonlinear microwave-dimer equation.

**Additional Case Study.**    To further evaluate RESTART on a challenging real scientific discovery setting beyond low-order algebraic laws, we add a second case study based on a nonlinear non-Hermitian microwave dimer model from recent experimental physics (Salcedo-Gallo et al., 2025):

$$n_{\mathrm{LC}}(\phi, G) \; = \; \frac{C_1^{G/C_2} \left( C_3 + \sin(\phi/C_4) \right) C_5 \; - \; C_6}{C_7}.$$

where $C_i$ are constants. The result is shown in Table 5 and Figure 16.

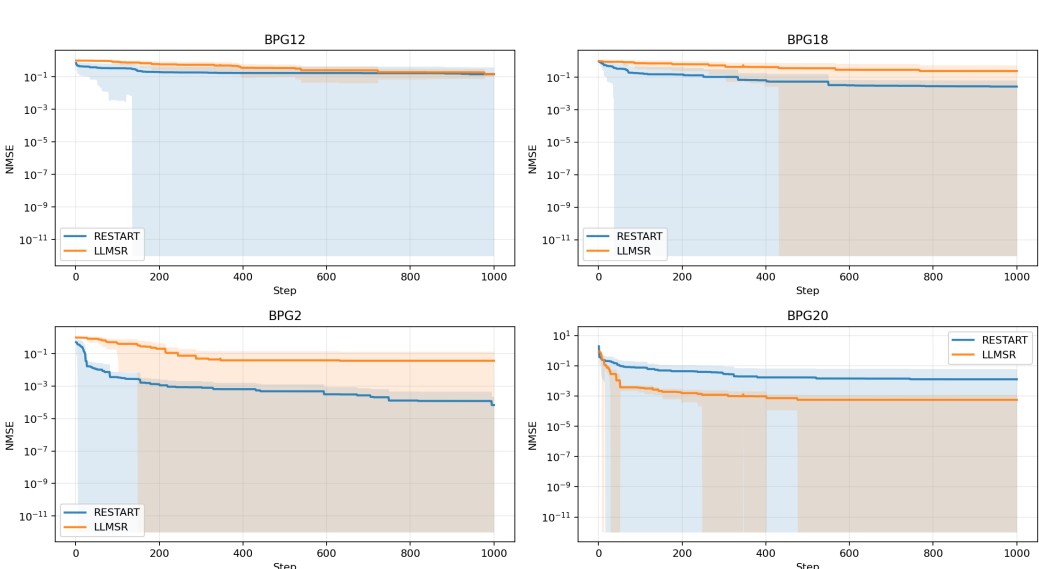

Figure 13: Average NMSE vs. Step on 4 Biology Equations, with shaded standard deviation.

## H  RESTART SEARCH PROCESS

Figure 17 compares the population growth rate equations discovered by RESTART and the baseline LLMSR. RESTART successfully recovers both key components of the ground-truth BPG12 equation: (i) the harmonic oscillation term $0.877\,P\sin(0.567t)$ capturing seasonal fluctuations, and (ii) the logistic growth term $0.701(1 - P/65.75)P$ with a realistic carrying capacity. In contrast, LLMSR produces a much noisier expression with only a weak logistic component and no harmonic interaction, yielding lower $R^2$ on both ID and OOD data.

To understand why RESTART succeeds, we inspect the prompt–response pair in Figure 18 and Figure 19. The prompt contains not only the current equations but also the top-ranked structures, two of which—`Harmonic Oscillator Interaction` and `Seasonal Variation with Damping`—are precisely the building blocks needed for the ground-truth solution.

This targeted guidance allows the LLM to synthesize a new hypothesis, `equation_v2`, that directly combines the harmonic interaction and logistic terms. As a result, the MSE dramatically decreases from 52.35 (for exemplar equation_v1) to $2.67 \times 10^{-9}$, effectively achieving near-perfect symbolic recovery. This case highlights how RESTART's iterative refinement and structure-guided prompting transform an initially rough approximation into a scientifically valid and highly accurate equation.

## I  PROMPT

Figure 2 illustrates the design of our adaptive prompt, which is dynamically assembled at each iteration of the hypothesize–experiment–analysis cycle. Unlike static few-shot prompting, our approach builds the prompt from three components that evolve over time:

- **Task Definition:** A fixed preamble that sets the LLM's role, clarifies the objective (finding an executable mathematical function), and specifies the variable description for given task.

- **Few-Shot Exemplars:** One or more previous hypotheses are inserted, along with their mean squared error, and the corresponding boosting equation analysis. These exemplars serve as in-context refinement guidance.

- **Accumulated Knowledge:** Finally, we include the structures from the structure library as reusable building blocks directly inserted into the comment section for next generation.

```python
def RESTART-1(Delta: np.ndarray, mass: np.ndarray, params: np.ndarray
    ) -> np.ndarray:

    # Step 1: Convert Delta from meV to Joules
    Delta_J = Delta * 1.602176634e-21 # 1 meV = 1.602176634e-21 J

    # Step 2: Base velocity using WKB approximation (m/s)
    velocity_base = np.sqrt(2 * np.abs(Delta_J) / mass)

    # Step 3: Convert velocity to km/s
    velocity_km_per_s = velocity_base / 1000

    # Step 4: Apply parameterized scaling
    velocity_km_per_s *= params[0]

    # Step 5: Phase-modulated nonlinear corrections
    # Phase-dependent sine and cosine corrections
    velocity_km_per_s += params[1] * np.sin(0.01 * Delta)
    velocity_km_per_s += params[2] * np.cos(0.001 * Delta)

    # Hyperbolic tangent correction with tunable phase
    phase_tanh = 0.1 + 0.05 * velocity_km_per_s
    velocity_km_per_s += params[3] * np.tanh(phase_tanh)

    # Nonlinear energy-dependent correction
    energy_term = np.sqrt(0.5 + 0.1 * Delta)
    velocity_km_per_s += params[4] * energy_term

    # Additional nonlinear transformation to stabilize variance
    # This term includes a tunable energy-dependent phase and
        amplitude
    phase_energy = params[5] * Delta + params[6]
    velocity_km_per_s += params[7] * np.sqrt(90.0 + 0.9 * np.abs(2.0 +
        30.0 * np.cos(phase_energy)))

    # Optional: Add a nonlinear feedback term to stabilize variance
    # This is an advanced term that can be used for more complex
        models
    if len(params) > 8:
        feedback_term = params[8] * np.log(1 + np.abs(Delta)) / (1 + np
            .abs(Delta))
        velocity_km_per_s += feedback_term

    return velocity_km_per_s
```

Figure 14: Discovered RESTART-1

```python
def RESTART-2(Delta: np.ndarray, mass: np.ndarray, params: np.ndarray
    ) -> np.ndarray:
    # Constants
    hbar = 1.0545718e-34
    eV_to_J = 1.602176634e-19 # 1 eV = 1.602e-19 J
    km_per_m = 1e-3 # 1 km = 1000 m

    # Convert Delta from meV to Joules
    Delta_J = Delta * 1e-3 * eV_to_J # meV to eV to J

    # Extract tunable parameters
    decay_rate = params[0] # B
    amplitude = params[1] # A
    oscillation_freq = params[2] # C
    phase_shift = params[3] # P
    quadratic_coeff = params[4] # E
    energy_scaling = params[5] # S
    velocity_scaling = params[6] # K
    offset = params[7] # D

    # Compute the exponential decay factor
    sqrt_Delta_J = np.sqrt(np.abs(Delta_J))
    decay_factor = np.exp(-decay_rate * sqrt_Delta_J)

    # Compute the oscillatory modulation
    oscillation = np.cos(oscillation_freq * sqrt_Delta_J + phase_shift
        )

    # Compute the quadratic correction term
    quadratic_term = 1 + quadratic_coeff * (Delta_J ** 2)

    # Compute the energy scaling factor
    energy_term = 1 + energy_scaling * Delta_J

    # Compute the velocity of probability flow
    velocity = (
        amplitude * decay_factor * oscillation * quadratic_term *
            energy_term *
        (sqrt_Delta_J / np.sqrt(mass)) * velocity_scaling * km_per_m
    ) + offset

    return velocity
```

Figure 15: Discovered RESTART-2

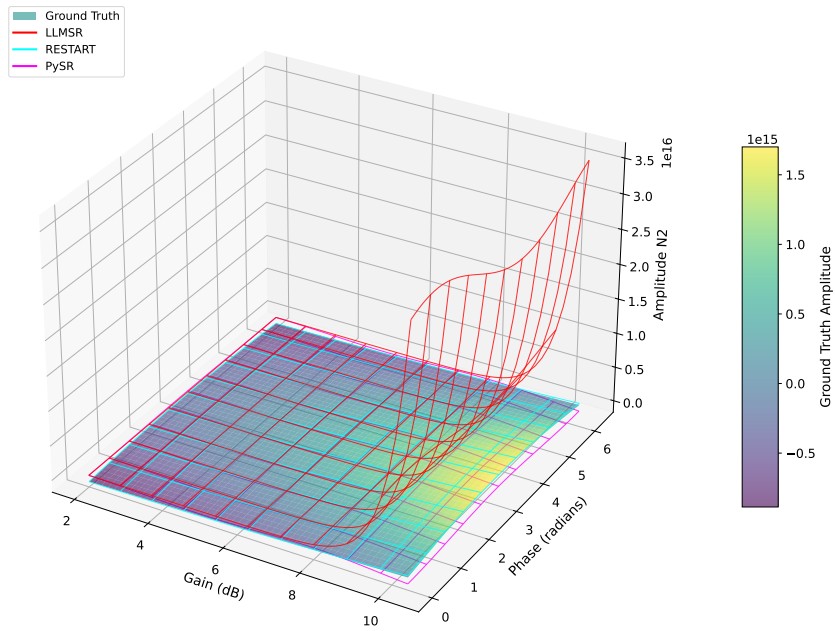

Figure 16: 3D comparison of amplitude trajectory under different methods.

```python
def RESTART_BPG12(t, P, params):
    r, a, omega, phi, K, A, theta, gamma, alpha, beta = params
    r_t = r * (1 + a * np.sin(omega * t + phi)) * np.exp(-gamma * t **
        beta)
    K_t = K * (1 + A * np.sin(omega * t + theta))
    harmonic = (np.sin(omega * t + phi) * P) * np.exp(-gamma * t ** beta)
    growth_rate = r_t * P * (1 - P / K_t) * alpha + harmonic * beta
    return growth_rate
```

```python
def llmsr_BPG12(t, P, params):
    r, K0, alpha, beta, eps, omega, eta, gamma, zeta, phi = params
    K = K0 + eta * t
    logistic = r * P * (1 - P / K)
    time_linear = alpha * t
    seasonal = eps * np.sin(omega * t)
    nonlinear = gamma * t ** 2
    delayed_P = np.roll(P, int(phi)); delayed_P[:int(phi)] = 0
    feedback = zeta * P * (1 - P / K) * np.sin(omega * (t - phi)) * (1 +
        0.1 * np.sin(omega * t))
    dP_dt = logistic + time_linear + beta + seasonal + nonlinear +
        feedback
    return dP_dt
```

Figure 17: Comparison of RESTART and LLMSR discovered functions for BPG12. The ground-truth equation is $f^\star(t, P) = 0.877\,P\sin(0.567t) + 0.701\,(1 - P/65.75)P$, a two-term structure combining a sinusoidal modulation and a logistic growth term. RESTART correctly recovers this structure, while LLMSR introduces additional linear, quadratic, and delayed-feedback components that deviate from the true form.

```
import numpy as np
MAX_NPARAMS = 10
params = [1.0]*MAX_NPARAMS

def equation_v0(...):
    """
    Analysis for current equation(f):
        Given (X,y), Current equation(f) has MSE: 52.35
        Boosted equation candidate: (((2.25 * (-1.50 + 0.0236*P)) *
            ...) ... )
        with MSE: 72.82
    """
    ...

def equation_v1(...):
    """
    Improved version of 'equation_v0'.
    Current equation(f) has MSE: 52.35
    Boosted equation candidate: (((0.994 * (0.078 + 0.044*current_pred
        )) * ...) ... )
    with MSE: 770.39
    """
    ...

# Top scoring concepts for the problem:
# - Harmonic Oscillator Interaction (fit: 95/100)
# - Damped Harmonic Interaction (fit: 88/100)
# - Seasonal Variation with Damping (fit: 85/100)
```

Figure 18: Partial prompt content used in the key iteration. It includes current loss, boosted residual equation, and top-ranked structural concepts.

```
def equation_v2(t: np.ndarray, P: np.ndarray, params: np.ndarray) ->
    np.ndarray:
    """
    Improved version of the population growth rate equation that
        incorporates:
    - Damped harmonic oscillation for seasonal variation
    - Logistic growth with time-dependent carrying capacity
    - Exponential damping
    - Harmonic interaction between population and time-dependent
        parameters
    """
    r, a, omega, phi, K, A, theta, gamma, alpha, beta = params
    r_t = r * (1 + a * np.sin(omega * t + phi)) * np.exp(-gamma * t **
        beta)
    K_t = K * (1 + A * np.sin(omega * t + theta))
    harmonic_interaction = (np.sin(omega * t + phi) * P) * np.exp(-
        gamma * t ** beta)
    growth_rate = r_t * P * (1 - P / K_t) * alpha +
        harmonic_interaction * beta
    return growth_rate
```

Figure 19: Partial LLM-generated response (equation_v2) that recovers the near-ground-truth BPG12 structure.

This step adapts the search to accumulated knowledge and focuses exploration on promising directions.

This dynamic construction allows RESTART to incorporate both short-term error feedback (via boosting equations) and long-term knowledge (via the structure library) into each prompt.

From the optimization perspective introduced in Section 2, this adaptive assembly can be viewed as an approximate solution to Eq. equation 2: instead of using a fixed prompt $p$ that may bias $q_p(\cdot)$ toward syntactically similar but suboptimal regions of the hypothesis space, we iteratively update $p$ by injecting exemplar feedback and discovered structures. This meta-optimization procedure progressively shifts the induced distribution $q_p$ toward regions of lower empirical loss, thereby increasing the probability that the best-of-$k$ sample achieves a smaller loss $\mathcal{L}(\tau)$.

