# OpenReview forum: "Robust Equation Structure Learning with Adaptive Refinement"
_ICLR.cc/2026/Conference — ICLR 2026 Poster_

### Official Review · Reviewer_APP1 · 2025-10-29

**Soundness:** 2
**Presentation:** 2
**Contribution:** 2
**Rating:** 4
**Confidence:** 2

**Summary:**

The paper proposes RESTART (Robust Equation STructure learning with Adaptive RefinemenT), a symbolic regression (SR) framework that explicitly closes the hypothesize → experiment → analyze loop.

In experiments, RESTART reports lower NMSE and higher symbolic accuracy than existing methods.

**Strengths:**

- The proposed method achieves better performance than existing LLM-based symbolic regression systems.

- The modular design that separates initialization, refinement, and structure retention could enable future extensibility.

**Weaknesses:**

Overall, this paper suffers from jargon-heavy and somewhat hand-wavy writing. Many parts need clearer, more concrete explanations and consistent notation. Below are detailed comments for revision:

Line 82: The term “structure proposal” is unclear. Please define explicitly what constitutes a structure—e.g., is it an equation template, code snippet, or symbolic expression tree?

Line 83: Define “equation exemplars.” Are these example equations used in few-shot prompts, or previously discovered hypotheses stored in a buffer?
Line 84: The notation for ( q_p ) and ( \tau_{1:k} ) is confusing. Replace ( \tau_{1:k} \sim q_p^k ) with the clearer and standard form ( \tau_i \sim q_p,\ i=1,\ldots,k ). Avoid the unconventional ( q_p^k ) notation.

Line 85: Explain “best-of-k” clearly—does it refer to selecting the lowest-loss sample among ( k ) generated equations?

Line 146: Remove the period in the section title. Also, “EFFICIENT INITIALIZATION” is vague—rename it to something more descriptive, such as “Initialization via Mapping-Based Estimator.”

Line 150: Specify what nonlinear and cross-variable dependencies are being modeled. Clarify what “using another estimator” means here—does the initialization merely provide a stronger approximation, or is it computationally more efficient?

Line 158: Equation (3) is syntactically incorrect. Replace ( g_t \in \arg\min_g L(g(f_t(x),x)) ) with the correct assignment form ( g_t \gets \arg\min_g L(g(f_t(x),x)) ). Additionally, clearly define ( t ) (iteration index) and ( g_t ) (the symbolic residual or exploration function). The following explanation seems incoherent—please rewrite it with precise mathematical meaning rather than generative phrasing.

Line 202: Clarify what ( L ) denotes (loss function) and what ( r^{f_t} ) refers to (is it the residual, ( y - f_t(x) )?).

Line 226: Variable names are inconsistent—change ( S_r ) → ( s_r ) and ( S_a ) → ( s_a ) to match prior definitions.

Line 231: Define what you mean by a “high-value structure.” Also, specify what the symbol h represents in the tuple ( (name, desc, h) ).

Line 295: Explain all symbols in the NMSE equation: ( y ) (true target), ( \bar{y} ) (mean of targets), and ( \hat{y} ) (predicted output). Introduce them before using the formula.

Line 359: The font size of Table 2 is excessively large. Remove unnecessary LaTeX sizing commands and ensure uniform table formatting.

Section 5.2.2: The example expression ( \sqrt{c_1 |x_1| / x_2} ) does not appear particularly challenging; classic search-based SR methods can likely recover it easily. Provide a stronger justification for why this case is nontrivial.

Line 450: In the expression “( i\Delta = E - V_0 + \hbar J_0 ),” remove the leading i—it appears to be a typographical error.

Figure 5a: Define both ( R^2(\text{data}) ) and ( R^2(\text{equation}) ) explicitly. Clarify how each is computed (e.g., is ( R^2(\text{data}) ) the fit to measured data and ( R^2(\text{equation}) ) the symbolic agreement with the analytical form?).

**Questions:**

Please try to address the concerns raised in the previous section and provide a revision of the paper during the rebuttal.

---

> ### Author Response · Authors · 2025-11-24
> **Rebuttal(1/2)**
>
> Thank you for the thoughtful and constructive review; we appreciate your insights and have addressed each concern in detail below:
>
> 1) **Line 82 — Define “structure proposal”.**
> We now avoid the overloaded term “structure” and replace it with equation template in the Problem Formulation section. An equation template $\tau \in \mathcal{T}$ is explicitly defined as a symbolic expression with fixed operators and free numeric constants, represented concretely as a concise Python code snippet.
>
> 2) **Line 83 — Define “equation exemplars”.**
> We explicitly define equation exemplars as concrete equation candidates discovered during search and stored in an exemplar buffer that conditions the adaptive prompting mechanism.
>
> 3) **Line 84 — Clarify sampling notation.**
> We adopt the  standard notation, $\tau_i \sim q_p,\ i=1,\ldots,k$. The objective now reads
> $\Phi_k(p)=\\mathbb{E}\_{\tau_i \sim q_p,\ i=1,\ldots,k}\left[\min_{1\le j\le k}\mathcal{L}(\tau_j)\right].$
>
> 4) **Line 85 — Define “best-of-$k$”.**
> We define best-of-$k$ loss as the selected loss of the best candidate among $k$ i.i.d. templates sampled from $q_p$.
>
> 5) **Line 146 — Rename “EFFICIENT INITIALIZATION”.**
> We rename the section to **Initialization via Transformer** for specificity and consistency with the method description.
>
> 6) **Line 150 — Clarify nonlinear/cross-variable dependencies.**
> We emphasize that the end-to-end Transformer initializer supplies a data-driven prior. We concretely enumerate nonlinearities (e.g., polynomial, trigonometric, exponential) and expand “cross-variable dependencies” to interaction terms between multiple variables. The ablation shows that stronger initialization improves accuracy.
>
> 7) **Line 158 — Fix Equation (3) syntax; define $t$ and $g_t$.**
> Let $t$ be the iteration index and $g:\mathbb{R}^{d+1}\to\mathbb{R}$ be the exploration (boosting-style) correction of hypothesis $f_t$ toward target $y$. We rewrite Eq. (3) with assignment and explicit arguments: $g_t \gets \arg\min_{g\in\mathcal{G}} \ \mathcal{L}\big(g(f_t(\mathbf{x}),\mathbf{x}),y\big).$
>
> 8) **Line 202 — Clarify $\mathcal{L}$ and residual notation.**
> We remove $r^{f_t}$. We now write losses explicitly as $\mathcal{L}\big(f(\mathbf{x}),y\big)$ and $\mathcal{L}\big(g(f_t(\mathbf{x}),\mathbf{x}),y\big)$ to eliminate ambiguity.
>
> 9) **Line 226 — Fix naming inconsistencies ($S_r\rightarrow s_r$ and $S_a\rightarrow s_a$).**
> We standardize notation and update the scoring equation to $s_{\text{fit}} = 100 \cdot \big(w_r s_r + w_a s_a\big).$
>
> 10) **Line 231 — Define “high-value structure” and the symbol $h$.**
> We define high-value templates as those with $s_{\text{fit}}\ge \alpha$, where $\alpha$ is a tunable threshold, and specify $h$ as the Python snippet (e.g., `np.sin(x)`) implementing the distilled template.
>
> 11) **Line 295 — Explain all symbols in the NMSE equation.**
> Before presenting NMSE, we introduce $y_i$, $\\hat{y}\_i$, $\\bar{y}$, $N_{\text{test}}$ (true target on index i, prediction on index i, mean of all true targets, and number of test samples, respectively).
>
> 12) **Line 359 — Fix large font in Table 2.**
> We remove ad-hoc LaTeX sizing.

---

> ### Author Response · Authors · 2025-11-24
> **Rebuttal(2/2)**
>
> 13) **Line 450 — Remove typo “$i\Delta$”.**
> We correct the expression to $\Delta = E - V_{0} + \hbar J_{0}$.
>
> 14) **Figure 5a — Define $R^2(\text{data})$ and $R^2(\text{equation})$.**
> We define $R^2(\text{data})$ as the $R^2$ computed against the measured ground-truth data points, and $R^2(\text{equation})$ as the $R^2$ computed on samples drawn from the reported analytical relation $v=\sqrt{\tfrac{2|\Delta|}{m}}$.
>
> ## Case Study: Case study equation seems simple—justify difficulty; include more complex example.
>
> While the simplified expression $\sqrt{c_1 |x_1| / x_2}$ appears elementary, the actual physical law used in the experiment is: $v=\sqrt{\frac{2|\Delta\cdot 1.602176634\times 10^{-22}|}{m}\Big/1000}$, which is more challenging. It involves (i) nested nonlinear operators (absolute value within a square root),  (ii) cross-variable interaction through division, and  (iii) large-magnitude unit-conversion constants embedded inside nonlinear transformations, which makes constant recovery and operator ordering difficult for search-based SR. The difficulty level is similar to the harder LLM-SRBench problems. We also reported the comparison with baseline method below.
>
> | Method | $R^2$(data) | $R^2$(equation) |
> |---|---:|---:|
> | RESTART-1 | **0.9827** | 0.9699 |
> | RESTART-2 | 0.9672 | **0.9985** |
> | LLMSR | 0.9688 | 0.9958 |
> | LLMDirect | 0 |
> | PySR | 0.8984 | 0 |
>
> Classic search-based SR (PySR) fails to reconstruct the absolute-square root structure and scalings, which only overfit on training data. On the other hand, only LLM-guided approaches succeed, and RESTART achieves the highest $R^2$ score. We also update Figure 6(b) for visual comparison.
>
> To further demonstrate that the methodology is not restricted to simple forms, we evaluated RESTART on a more complex scientific equation[1] involving trigonometric interactions, exponential terms, and constants spanning over 20 orders of magnitude:
> $n_{\mathrm{LC}}(\phi,G) =\frac{C_1^{G/C_2}\bigl(C_3 + \sin(\phi/C_4)\bigr)C_5-C_6}{C_7}$
>
> | Method | $R^2$ |
> |---|---:|
> | RESTART | 0.9771 |
> | LLMSR | 0.1737 |
> | LLMDirect | 0 |
> | PySR | 0.5325 |
>
> RESTART attains near-perfect accuracy, while both LLMSR and PySR degrade significantly. This demonstrates that RESTART’s identify$\rightarrow$correct refinement is not limited to simple forms and provides robust performance on structurally intricate, physically meaningful equations.
>
> The Qwen3-8B model used in our experiments was released on April 28, 2025, prior to the paper release date, which rules out the data leakage issue.
>
> | Item | Public date |
> |---|---|
> | Qwen3-8B release | 28 Apr 2025 |
> | Energy–speed Equation [2] | 02 Jul 2025 |
> | Amplitude Equation[1] | 05 Aug 2025 |
>
> We have also revised the manuscript to explain in detail why the equation is non-trivial.
>
> [1] Demonstration of a Tunable Non-Hermitian Nonlinear Microwave Dimer. Nature Communications 16, no. 1 (2025): 7193. https://doi.org/10.1038/s41467-025-62620-1
>
> [2] Violetta Sharoglazova, Marius Puplauskis, Charlie Mattschas, Chris Toebes, and Jan Klaers. Energy–speed relationship of quantum particles challenges Bohmian mechanics. Nature, 643 (8070):67–72, July 2025. ISSN 1476-4687. doi: 10.1038/s41586-025-09099-4. URL https: //doi.org/10.1038/s41586-025-09099-4.

---

> ### Author Response · Authors · 2025-11-27
>
> Dear Reviewer APP1,
>
> Thank you again for your careful reading and concrete suggestions on the writing. In our latest revision, we refined the manuscript to make it more formal and clear, corrected the typos and issues you noted, and expanded the case study section to better explain its difficulty and to include a more complex example with baseline comparisons. We hope these revisions address your concerns, and we’re happy to make further adjustments if you have additional questions.

---

### Official Review · Reviewer_bYpV · 2025-11-01

**Soundness:** 3
**Presentation:** 3
**Contribution:** 2
**Rating:** 4
**Confidence:** 4

**Summary:**

This paper introduces RESTART, an LLM-based symbolic regression framework that combines LLM-based hypothesis generation with several guidance mechanisms: (1) efficient initialization, (2) short-term refinement via boosting-style residual analysis, and (3) long-term knowledge retention through a structure library. Experiments on LLM-SRBench show better performance compared to baselines including LLMSR, with good efficiency gains.

**Strengths:**

* The framework makes intuitive sense: fitting symbolic functions to residuals provides data-driven feedback rather than generic error signals, and the structure library of motifs offers a principled way to accumulate good patterns.
* Good evaluation setup and empirical results on LLM-SRBench problems. The efficiency analysis showing comparable performance with 25% of iterations is valuable. Figures are well-designed and good ablation studies are provided in the main paper and the appendix.
* The writing is clear, literature coverage is reasonable, and the appendix provides useful implementation details. The problem motivation around completing the scientific discovery cycle is well-articulated.

**Weaknesses:**

* Limited novelty: The contribution consists of fairly straightforward additions to the LLM-SR framework: pre-trained initialization, residual fitting for guidance, and a pattern library. While sensible, these feel incremental: residual analysis resembles gradient boosting adapted to SR, and structure libraries conceptually overlap with recent concept library approaches, though implemented differently.
* Residual fitting assumption: The boosting formulation fits g(x) to the residual (y - f_t(x)), which appears to assume the missing component is additively separable from the current hypothesis. This may not hold when the true structure involves multiplicative or other nonlinear compositions with respect to f_t(x). See questions for details.
* Evaluation gaps: Only numeric accuracy metrics are reported (NMSE, Acc_threshold), not symbolic correctness. Variation across problems isn't visualized beyond 4 selected examples. The case study lacks baseline comparisons and doesn't provide the prompt used. Several design choices (fitness scoring, ablation scope) need better justification. See questions for specifics.

**Questions:**

1. Most SR backends like E2E don't condition on f_t(x) form. Is g actually receiving f_t(x) or just fitting x to the residual? What exactly does each subproblem backend receive?

2. How does the method handle cases where the missing structure isn't additively separable (e.g., f*(x) = h(x) · k(x))? Can you provide benchmark examples and show if performance degrades on such problems? Intuitively, the residual model would not work well in such cases and would suggest functions that are not appropriate. This might guide the framework towards predicting more complex linear forms (potentially with nonlinear basis functions). To be clear, even with this limitation, the additional component could be valuable in many use cases, but it would be helpful to discuss this in the paper and provide analysis on the effect on final discovered forms.

3. Can you report symbolic accuracy metrics (e.g., symbolic recovery like in LLM-SRBench) alongside numeric error? Can you show performance distributions across all problems rather than just aggregated means?

4. For the case study: what was the exact prompt used? How do baselines like LLM-SR perform on the same problem? The problem is published in 2025, after the cutoff date of many LLMs, but how do you rule out that the LLM is recalling or deriving the simple equation form rather than discovering it?

5. What does "Additive" mean in Figure 4? Why are the ablations conducted only on 2 problems, and how are they selected? Is there an ablation on the fitness score design (absolute, relative, and weighted s_fit)?

Suggestions:
* I would suggest authors to bring examples of prompts and how the two mechanisms work to the main paper for clarity.
* I was wondering if the authors could provide the code for reproducibility and evaluation.
* minor: some typos exist in the paper (for example: "Libarary -> "Library" in Figure 1)

---

> ### Author Response · Authors · 2025-11-24
> **Rebuttal(1/4)**
>
> Thank you for the thoughtful and constructive review; we appreciate your insights and have addressed each concern in detail below:
>
> ## Weakness 1: Limited novelty relative to LLM-SR / boosting / concept libraries
>
> Our contribution is not a new SR solver, but a new framework in the SR discovery loop: an explicit "identify $\rightarrow$ revise" stage that symbolically diagnoses structural error and uses that diagnosis to guide LLM revision. Prior LLM-based SR methods largely implement "hypothesize $\rightarrow$ experiment" and feed back only weak signals (scalar loss, raw residuals, or brief text), leaving the LLM to generate new hypotheses with potentially lower error. RESTART instead:
>
> 1. Formulates a targeted symbolic subproblem $g_t  \gets arg\min_{g} \mathcal{L}(g(f_t(x), x),y),$ that explicitly models the structure from $f_t$. This makes the LLM a guided reviser conditioned on explicit structural corrections, not a blind sampler.
>
> 2. Unlike earlier concept libraries that store natural-language hints [1] and require noisy grounding at generation time, RESTART distills validated refinements into structural priors as code snippets that can be directly composed into future hypotheses, which eliminates the NL $\rightarrow$ symbolic translation gap. Additionally, the structural library retains only fitness-gated improvements to ensure retained structures are important for equation discovery.
>
> Additionally, the empirical results show that RESTART outperformed the baselines in LLM-SRBENCH through its novel analysis stage.
>
> ## Weakness 2: “Additive residual” assumption
>
> We appreciate this important point; however, it stems from a misunderstanding of Eq. (3), $g_t \gets \arg\min_{g} \mathcal{L}(g(f_t(x), x),y)$. We have revised the manuscript for more detailed definition of $g_t$ to avoid confusion.
>
> 1. Clarifying the boosting formulation:
>
>     we do *not* fit $g(x) = residual = y - f_t(x)$.
>         RESTART does *not* use the classical residual boosting form $g(x) = residual = y - f_t(x).$ Instead, the exploration subproblem is defined as $g_t : \mathbb{R}^{d+1} \to \mathbb{R}, \quad
>     g_t(f_t(x), x) = y,$ where
>     - $f_t \colon \mathbb{R}^d \to \mathbb{R}$ is the current symbolic hypothesis, and
>     - $g_t$ receives both the scalar prediction $f_t(x)$ and the original input $x \in \mathbb{R}^d$ as its arguments.
>
>     In other words, the subproblem backend is always given the data $\{((f_t(x_i),x_i),y_i)\}^N_{i=1}$. The value $f_t(x_i)$ is treated as an additional feature, so the solver learns a function over $(f_t(x), x)$, not a function over $x$ that tries to approximate the residual $y - f_t(x)$.
>
> 2. Why no additive separability assumption is required:
>
>     Because $g_t$ operate on $(f_t(x), x)$, it can represent arbitrary compositions between the current hypothesis and the input, for example:
>     - multiplicative corrections, e.g. $g_t(f_t(x), x) = f_t(x) \cdot x$,
>     - nonlinear warping, e.g. $g_t(f_t(x), x) = \sin(f_t(x))$,
>     - rational structures, e.g. $g_t(f_t(x), x) = \frac{f_t(x)}{1 + h(x)}$.
>
>     Thus, RESTART does **not** assume that the missing structure is additively separable from $f_t(x)$. In other words, RESTART targets structural gap modeling, not additive residual boosting.
>
> 3. Additive-residual boosting appears as an ablation (“Additive” in Fig. 4).
>
>     The label “Additive” in Fig. 4 refers specifically to an ablation where the exploration function behaves like classical residual boosting by fitting
>     $
>     \tilde{g}(x) = y - f_t(x),
>     $
>     and then combining $f_t(x)$ and $\tilde{g}(x)$ additively. This ablation was included precisely to test your concern about additive residual assumption.
>
>     |      | NMSE      |
>     |-------------|---------|
>     | RESTART   | 3.499e-07 |
>     | Additive  | 9.886e-04 |
>
>     As shown in the ablation results, this “Additive” variant performs substantially worse than the full RESTART formulation that learns $g_t(f_t(x), x)$. This empirically confirms that:
>     - the gains do not come from simple additive residual fitting, and
>     - conditioning on $f_t(x)$ within a more general function $g_t(f_t(x), x)$ is crucial for the observed improvements and robustness.

---

> ### Author Response · Authors · 2025-11-24
> **Rebuttal(2/4)**
>
> ## Weakness 3: Evaluation gaps (symbolic accuracy, distributions, case-study details)
>
> 1. Symbolic Accuracy
>
>     We agree that symbolic accuracy is essential to our method, however current LLM-as-judge SA metric defined in LLM-SRBench measures syntactic equivalence between the discovered expression and a single canonical ground-truth form provided by the benchmark. In contrast, RESTART’s objective is to recover an equation that best explains the data and underlying dynamics, regardless of whether its symbolic tree matches the exact canonical template.  The key point is that SA measures syntactic equivalence, not functional correctness. RESTART yields substantial improvements in NMSE, $\text{ACC}_{0.1}$, but its Symbolic Accuracy is below the LLMSR baseline. We acknowledge this observation and explain below why it is an anticipated consequence of the algorithmic design.
>
>     | Method      | LSR-Transform | Biology | Material Science | Physics |
>     |-------------|--------------:|--------:|-----------------:|--------:|
>     | LLMSR       | **47.30**     | 1.53    | 3.52         | **10.23** |
>     | RESTART     | 34.83         | 2.55| 1.80             | 6.82 |
>     | E2E         | 9.01          | 0.00    | **8.00**         | 0.00 |
>     | PySR        | 37.84         | **4.17**| 4.00             | 9.09 |
>     | SGA         | 9.91          | 2.71    | 3.13             | 2.46 |
>     | LlmDirect   | 13.06         | 1.56    | 6.50             | 1.42 |
>
>
>     The potential reason is that RESTART is built around an explicit Identify $\rightarrow$ Correct loop. At each iteration we construct a symbolic exploration function $g_t(f_t(x), x)$ and use it to diagnose structural deficiencies in the current hypothesis. This process induces a sequence of nonlinear, compositionally rich refinements, some of which remain in the final expression. As a result, Iterative structural refinement produces semantically correct, but syntactically different, expressions. Even though the structure library provides guidance from a limited set, the final expression could still diverge from the ground truth in form but numerically close. Thus, RESTART’s tendency to introduce additional structurally meaningful factors reduces SA even when numerical behavior is essentially perfect. Two expressions that are algebraically compatible or even empirically indistinguishable receive an SA score of zero if their symbolic trees differ.
>
>     As an example, the ground-truth BPG20 equation is a combination of logistic interaction terms and a sigmoidal correction:
>
>     $0.139(-1 + \frac{P(t)}{8.038})(1 - \frac{P(t)}{70.018})P(t) + 0.139(1 - \frac{P(t)}{70.018})P(t) + \frac{0.139P(t)}{1 + \exp(-8.038(-0.589 + P(t)))}$
>
>     RESTART recovers the underlying dynamics but expresses them through a richer multiplicative structure that reflects the iterative corrections inferred during refinement:
>
>     $G(t)=\left[C_0 P(t)\left(1-\frac{P(t)}{C_1}\right)\right]\left(1+\sin(C_2 t + C_3)+\cos(C_4 t+C_3)e^{-C_5 t}+ C_7 P(t)\sin(C_2 t+C_3)\right)\cdot e^{-C_6 t} e^{-C_9 P(t)}\left(1 + C_8\sin(C_2 t+C_3)\right).$
>
>     After fitting constants, the two expressions are nearly indistinguishable:
>
>     - ID $R^2$ = 0.999
>     - OOD $R^2$ = 0.999
>     - ID NMSE = 1.704e-07
>     - OOD NMSE = 8.586e-07
>
>     Figure 3 (in the paper) shows that the curves coincide to the plotting resolution. Nevertheless, because the discovered symbolic structure is not syntactically identical to the canonical template, the SA metric marks it as incorrect.
>
>     RESTART’s primary contribution is the introduction of an analysis-driven corrective loop that enhances numerical performance across scientific domains. This mechanism does not enforce symbolic canonicalization, and thus SA may be lower even when the functional recovery is substantially stronger.
>
> 2. Distributions across tasks
>
>     We include standard deviations in Tables 1–2, and an additional box plot for distribution demonstration(As shown in Appendix C, Figure 7).

---

> > ### Author Response · Authors · 2025-11-24
> > **Rebuttle(3/4)**
> >
> > ## Weakness 3(continued)
> >
> > 3. Case study prompt and baselines.
> >
> >      The case study uses the same standardized problem template as all tasks in Fig. 13.  The task-specific portion of the prompt is:
> >
> >         "You are a helpful assistant tasked with discovering mathematical function structures for scientific systems.
> >
> >         Complete the 'equation' function below, considering the scientific meaning and relationships of inputs.
> >         Find the mathematical function skeleton that represents speed: velocity of probability flow in the classically forbidden region, expressed in km/s, given data on Delta: energy difference $\\Delta = E - V_{0} + \\hbar J_{0}$ expressed in meV, and mass: particle mass in kg."
> >
> >     This prompt supplies only a semantic description of the quantities and does not provide the mathematical form of the underlying physical relation.
> >
> >     To ensure fairness, RESTART, LLMSR, and LLMDirect were given the same task specific prompt.
> >     LLMDirect is an ablation where the LLM attempts to propose a valid equation directly from the background description, without iterative refinement or analysis.
> >
> >     | Method        | $R^2$(data) | $R^2$(equation) |
> >     |---------------|----------|--------------|
> >     | RESTART-1     | **0.9827**   | 0.9699       |
> >     | RESTART-2     | 0.9672   | **0.9985**   |
> >     | LLMSR         | 0.9688   | 0.9958       |
> >     | PySR          | 0.8984   | 0            |
> >     | LLMDirect     | 0        | 0            |
> >
> >     Classic search-based SR (PySR) fails to reconstruct the absolute-squreroot structure and scalings, which only overfit on training data. LLMDirect also fails, indicating that the model cannot recover the equation from recalling background knowledge alone and that the iterative identify$\rightarrow$correct loop is essential.
> >
> >     To rule out contamination from the 2025 Nature publication:
> >
> >     1. **Model cutoff dates.**
> >     The Qwen3-8B model used in our experiments was released on April 28, 2025, prior to the report release date, which rules out the data leakage issue.
> >
> >         | Item | Public date |
> >         |---|---|
> >         | Qwen3-8B release | 28 Apr 2025 |
> >         | Energy–speed Equation [2] | 02 Jul 2025|
> >
> >     2. **LLMDirect failure.**
> >     If the LLM had memorized or recalled the closed-form relation, LLMDirect would recover it from the textual prompt alone. Instead, it produces incorrect, low-$R^2$ expressions, demonstrating that the LLM cannot retrieve this equation from prior knowledge.
> >
> >     Together, these observations indicate that RESTART is discovering the relation via iterative reasoning rather than recalling it.
> >
> > 4. Ablation study on hyperparameters
> >
> >     We added an additional sensitivity study over two equations in the Biology Domain (BPG18, BPG20). We vary the fitness gate $\alpha$ from 40 $\rightarrow$ 70 and shifting between relatively-weighted and absolutely-weighted as listed below.  The result yields only small changes in NMSE across settings, which shows the hyperparameter robustness under different settings.
> >
> >
> >     | Setting Type          |  $\alpha$ | \(w_r\) (Relative Weights)         | \(w_a\) (Absolute Weights)        |
> >     |-----------------------|-------------------------------|------------------------------------|------------------------------------|
> >     | Default           | 40                            | [0.3, 0.5, 0.8, 1.0]                | [0.7, 0.5, 0.2, 0.0]                |
> >     | Rel Weighted      | 40                            |  [0.5, 0.7, 0.9, 1.0]                | [0.5, 0.3, 0.1, 0.0] |
> >     | Abs Weighted      | 40                            |  [0.1, 0.3, 0.5, 0.7]                | [0.9, 0.7, 0.5, 0.3] |
> >     | $\alpha = 50$     | 50                            | [0.3, 0.5, 0.8, 1.0]                | [0.7, 0.5, 0.2, 0.0]                |
> >     | $\alpha = 70$     | 70                            | [0.3, 0.5, 0.8, 1.0]                | [0.7, 0.5, 0.2, 0.0]                |
> >
> >
> >     |  | BPG18 | BPG20 |
> >     |--------|-----------:|-----------:|
> >     | Default | 2.25e-02 | 1.70e-07 |
> >     | $\alpha = 50$ | 2.14e-02 | 2.13e-06 |
> >     | $\alpha = 70$  | 1.60e-02 | 3.77e-08 |
> >     | Abs Weighted | 2.02e-02 | 8.31e-06 |
> >     | Rel Weighted | 1.24e-02 | 1.15e-07 |
> >
> >
> > ## Question 1: What exactly does each subproblem backend receive?
> >
> > The subproblem solver receives data $\{((f_t(x_i),x_i), y_i)\}^N_{i=1}$, to solve   $g_t \gets arg\min_{g} \mathcal{L}(g(f_t(x), x),y)$, Please refer to weakness 2 for more details.
> >
> > ## Question 2: Additive Assumption
> >
> > Please refer to Weakness 2.

---

> > > ### Author Response · Authors · 2025-11-24
> > > **Rebuttal(4/4)**
> > >
> > > ## Question 3: Symbolic Accuracy
> > >
> > > Please refer to Weakness 3.1.
> > >
> > > ## Question 4: Case Study
> > >
> > > Please refer to Weakness 3.3.
> > >
> > > ## Weakness 5: Meaning of “Additive” in ablation
> > >
> > > Please refer to Weakness 2. We revised the manuscript to include a discussion about additive ablation to avoid misunderstanding. The ablation study is performed on BPG12, which is randomly chosen from the 4 challenging equations in the Biology Domain.
> > >
> > > ## Suggestion: Add prompt examples in main text.
> > >
> > > We moved the prompt example from appendix to the main text.
> > >
> > > ## Suggestion 2: Release code after acceptance.
> > >
> > > We will release the code after acceptance.
> > >
> > > ## Suggestion 3: Fix typos (e.g., “Libarary $\rightarrow$ Library”).
> > >
> > > Thank you for pointing this out, we have revised the manuscript.
> > >
> > > [1] Grayeli, Arya, Atharva Sehgal, and Omar Costilla-Reyes. Symbolic Regression with a Learned Concept Library. n.d.
> > > [2] Violetta Sharoglazova, Marius Puplauskis, Charlie Mattschas, Chris Toebes, and Jan Klaers. Energy–speed relationship of quantum particles challenges Bohmian mechanics. Nature, 643 (8070):67–72, July 2025. ISSN 1476-4687. doi: 10.1038/s41586-025-09099-4. URL https: //doi.org/10.1038/s41586-025-09099-4.

---

> ### Author Response · Authors · 2025-11-27
>
> Dear Reviewer bYpV,
>
> Thank you again for your detailed and constructive feedback. In our previous response, we clarified what is new about RESTART relative to prior approaches, explained why our formulation does not rely on an additive-residual assumption and reported the “Additive” ablation, expanded our discussion of symbolic accuracy and performance distributions, provided more details on the case study, added hyperparameter ablations, moved the prompt examples into the main text, and fixed the figure typo you pointed out. We hope these resolved your concerns, and if anything still needs further clarification, we would be glad to address it.

---

### Official Review · Reviewer_NteJ · 2025-11-02

**Soundness:** 3
**Presentation:** 3
**Contribution:** 3
**Rating:** 6
**Confidence:** 5

**Summary:**

This paper proposes RESTART, a robust large-language-model-based framework for symbolic equation discovery that integrates principled short-term and long-term analytical guidance into the discovery process. The authors evaluate RESTART on 239 problems from the recent LLM-SRBench benchmark and report substantial improvements over the state-of-the-art baseline LLM-SR. The paper is generally well-motivated, clearly written, and supported by extensive experimental evaluation and insightful analyses.

**Strengths:**

- Although individual ideas (boosting-style refinement, structure library) are not novel alone, their integration into the discovery framework is novel and seem to provide considerable performance improvement.
- Experiments on a wide range of datasets from LLM-SRBench are thorough and demonstrate consistent gains over state-of-the-art baselines.
- The paper includes comprehensive ablation studies and a meaningful real-world physics case study which strengthen their evaluation.
- The paper is generally very well written and easy to follow.

**Weaknesses:**

- I noticed that authors are only reporting results on 93 out of 128 LSR-Synth from the llm-srbench benchmark. why is that and what criteria is behind this selection of 93 out of 128 problems?

- In pg 6, Acc₀.₁ is described as the symbolic accuracy from [1], but it is actually a stricter numeric metric rather than the symbolic accuracy (SA), which in [1] is computed via an LLM-as-judge for symbolic equivalence. I would suggest authors to also report SA in their results (see fig. 11 in [1]).

- The analysis in case study of sec 5.2.2 on real experimental data is insightful, however, with the current report of results it's not clear what are the symbolic forms of equation discovered by RESTART and how's the advantage of these new hypothesis compared to the ones from LLM-SR baseline. Providing symbolic forms similar to Figure 12 for some examples or at least this case study could be helpful to better understand the problem.

- Some abrupt jumps in Fig. 4a suggest high variance in the observed results. Are these from a single run or averaged over multiple runs? Can you also report the confidence of these curves across different runs?

- Authors have not shared to the anonymous version of code in their submission. It would be good if authors can also share their code if paper is accepted so that research community can also build on it.

**Minor Comment:**
- It would be helpful to include the LLM backbone model in the captions of Tables 1 and 2 for clarity.

[1] LLM-SRBench: A New Benchmark for Scientific Equation Discovery with Large Language Models, ICML 2025

**Questions:**

check weaknesses section

---

> ### Author Response · Authors · 2025-11-24
> **Rebuttal(1/2)**
>
> Thank you for the thoughtful and constructive review; we appreciate your insights and have addressed each concern in detail below:
>
> ## Weakness 1: Report 3 out of 4 domains from LLM-SRbench
>
> Due to computational limitations, we only choose 3 out of 4 domains from LSR-Synth. We further test on the first 6 equations of the Chemistry dataset. We achieve SOTA on 5 equations in-domain and 4 equations out-of-domain.
>
> ### In-Distribution
> | method| CRK0| CRK1| CRK2 | CRK3| CRK4 | CRK5|
> |-----------|------------|-------------|-------------|-------------|-------------|-------------|
> | **RESTART** | **3.000e-06** | 2.772e-07  | **1.990e-06** | **9.963e-05** | **7.098e-06** | **8.853e-08** |
> | E2E| 0.01828| 0.2603| 0.001105| 0.7304| 0.04842| 0.01069|
> | LlmDirect | 0.002733| 9.953e-05   | 0.7147| 0.04499| 0.06074| 0.002380|
> | LLMSR| 5.904e-06  | **5.898e-09** | 2.524e-06| 9.989e-05| 1.965e-05| 3.830e-07|
> | PySR | 1.276e-04  | 6.914e-07   | 1.402e-05|3.973e-04   | 0.00435| 7.104e-04|
> | SGA| 2.747e-04| 5.223e-05   | 0.02458| 2.768e-04   | 0.003115| 0.02674|
>
> ### Out-of-Distribution
> | method| CRK0| CRK1| CRK2| CRK3| CRK4| CRK5|
> |-----------|-------------|--------------|-------------|-------------|-------------|--------------|
> | **RESTART** | **9.015e-04** | 0.003233     | 0.003014| **0.9215**  | **0.2433**  | **4.311e-04** |
> | E2E       | 5.527       | 1.343        | 0.1172      | 11.96 | 369.65      | 0.7392       |
> | LlmDirect | 3.189       | 0.05137      | 271.14      | 2578.76| 2589.51| 1.373        |
> | LLMSR     | 0.006033    | 4.176e-06    | 0.001522    | 1.482 | 1.054  | 8.193e-04    |
> | PySR      | 0.005010    | **4.102e-06** | **1.602e-04** | 2.612| 55.94    | 0.2464       |
> | SGA       | 0.04419     | 1.152        | 5.540       | 1.897       | 55.54 | 8.820        |
>
> ## Weakness 2: Inclusion of Symbolic Accuracy
>
> We will revise the manuscript to avoid the confusion. We agree that symbolic accuracy is essential to our method, however current LLM-as-judge SA metric defined in LLM-SRBench measures syntactic equivalence between the discovered expression and a single canonical ground-truth form provided by the benchmark, not functional correctness. RESTART yields improvements in NMSE, $\text{ACC}_{0.1}$, but its Symbolic Accuracy is below the LLMSR baseline.
>
> | Method| LSR-Transform | Biology | Material Science | Physics |
> |-------------|--------------:|--------:|-----------------:|--------:|
> | LLMSR| **47.30** | 1.53    | 3.52 | **10.23** |
> | RESTART | 34.83  | 2.55| 1.80  | 6.82 |
> | E2E | 9.01 | 0.00    | **8.00**  | 0.00 |
> | PySR | 37.84 | **4.17**| 4.00  | 9.09 |
> | SGA | 9.91 | 2.71    | 3.13  | 2.46 |
> | LlmDirect   | 13.06| 1.56    | 6.50 | 1.42 |
>
> The potential reason is that RESTART is built around an explicit Identify $\rightarrow$ Correct loop. At each iteration we construct a symbolic exploration function $g_t(f_t(x), x)$ and use it to diagnose structural deficiencies in the current hypothesis. This process induces a sequence of nonlinear, compositionally rich refinements, some of which remain in the final expression. As a result, Iterative structural refinement produces semantically correct, but syntactically different, expressions. Even though the structure library provides guidance from a limited set, the final expression could still diverge from the ground truth in form but numerically close. Thus, RESTART’s tendency to introduce additional structurally meaningful factors reduces SA even when numerical behavior is essentially perfect. Two expressions that are algebraically compatible or even empirically indistinguishable receive an SA score of zero if their symbolic trees differ.
>
> As an example, the ground-truth BPG20 equation is a combination of logistic interaction terms and a sigmoidal correction:
>
> $0.139(-1 + \frac{P(t)}{8.038})(1 - \frac{P(t)}{70.018})P(t) + 0.139(1 - \frac{P(t)}{70.018})P(t) + \frac{0.139P(t)}{1 + \exp(-8.038(-0.589 + P(t)))}$
>
> RESTART recovers the underlying dynamics but expresses them through a richer multiplicative structure that reflects the iterative corrections inferred during refinement:
>
> $G(t)=\left[C_0 P(t)\left(1-\frac{P(t)}{C_1}\right)\right]\left(1+\sin(C_2 t + C_3)+\cos(C_4 t+C_3)e^{-C_5 t}+ C_7 P(t)\sin(C_2 t+C_3)\right)\cdot e^{-C_6 t}e^{-C_9 P(t)}\left(1 + C_8\sin(C_2 t+C_3)\right).$
>
> After fitting constants, the two expressions are nearly indistinguishable:
>
> - ID $R^2$ = 0.999
> - OOD $R^2$ = 0.999
> - ID NMSE = 1.704e-07
> - OOD NMSE = 8.586e-07
>
> Figure 4 shows that the curves coincide to the plotting resolution. Nevertheless, because the discovered symbolic structure is not syntactically identical to the canonical template, the SA metric marks it as incorrect.
>
> RESTART’s primary contribution is the introduction of an analysis-driven corrective loop that enhances numerical performance across scientific domains. This mechanism does not enforce symbolic canonicalization, and thus SA may be lower even when the functional recovery is substantially stronger.

---

> ### Author Response · Authors · 2025-11-24
> **Rebuttal(2/2)**
>
> ## Weakness 3: Symbolic form for case study
>
> We add the program forms discovered (RESTART1 and RESTART2) to Appendix G, and report them in the equations below:
>
> $\begin{aligned}
> RESTART1(\Delta,m,p)&=\frac{p_0}{1000}\sqrt{\frac{2\left|\Delta(1.602176634\times10^{-21})\right|}{m}}+ p_1\sin(0.01\Delta)+ p_2\cos(0.001\Delta) \\\\
> &\quad+ p_3\tanh\left(0.1 + 0.05\left[\frac{p_0}{1000}\sqrt{\frac{2\left|\Delta(1.602176634\times10^{-21})\right|}{m}}+ p_1\sin(0.01\Delta)+ p_2\cos(0.001\Delta)\right]\right) \\\\
> &\quad+ p_4\sqrt{0.5 + 0.1\Delta}+ p_7\sqrt{90 + 0.9\left|2 + 30\cos(p_5\Delta + p_6)\right|}+ p_8\frac{\ln(1+|\Delta|)}{1+|\Delta|}
> \end{aligned}
> $
>
> $\begin{aligned}
> RESTART2(\Delta,m;\mathbf p) &= p_{1}e^{-p_{0}\sqrt{|\Delta_J|}}\cos\bigl(p_{2}\sqrt{|\Delta_J|}+p_{3}\bigr)\left(1+p_{4}\Delta_J^{2}\right)\left(1+p_{5}\Delta_J\right)\frac{\sqrt{|\Delta_J|}}{\sqrt{m}}p_{6}(10^{-3})+ p_{7},\\\\
> \Delta_J &= 1.602176634\times10^{-22}
> \end{aligned}$
>
> and we also include the comparison against other baselines:
>
> | Method        | $R^2$(data) | $R^2$(equation) |
> |---------------|----------|--------------|
> | RESTART-1     | **0.9827**   | 0.9699       |
> | RESTART-2     | 0.9672   | **0.9985**   |
> | LLMSR         | 0.9688   | 0.9958       |
> | PySR          | 0.8984   | 0            |
> | LLMDirect     | 0        | 0            |
>
> ## Weakness 4: Efficency Trajectory
>
> Following LLMSR, which reports the best trajectory, we report the trajectory of the median run to better represent the result. We will now include the mean result and standard deviation in the Appendix F. As shown in Figure 13, we reported the average NMSE trajectories for four biology equations. Note that because NMSE values differ on a log scale, the mean can be dominated by the higher-NMSE trajectories. For example, if there are 10 trajectories: one with NMSE $1\times10^{-4}$ and nine with NMSE $1\times10^{-9}$; the average NMSE will still be above $1\times10^{-5}$, which is substantially higher than most of the trajectories.
>
> ## Weakness 5: Open Source
>
> We will release the code after the acceptance.
>
> ## Minor Comments
>
> Thank you for pointing this out. We have indicated that the experiments were conducted using Qwen3-8B [1].
>
> [1] Qwen Team. Qwen3 technical report, 2025. URL https://arxiv.org/abs/2505.09388.

---

> ### Author Response · Authors · 2025-11-27
>
> Dear Reviewer NteJ,
>
> Thank you again for your thoughtful review and valuable comments. In our previous response, we expanded our experiments to include the first 6 equations in the Chemistry domain from LLM-SRBench, provided and discussed the symbolic accuracy results from RESTART, added the explicit symbolic forms discovered for the case study, and included averaged trajectories. We hope this addresses your concerns; if you have additional comments, we would greatly appreciate your feedback and are happy to address your concern.

---

### Official Review · Reviewer_PQgs · 2025-11-03

**Soundness:** 3
**Presentation:** 2
**Contribution:** 2
**Rating:** 6
**Confidence:** 5

**Summary:**

The paper “Robust Equation Structure Learning with Adaptive Refinement (RESTART)” presents a novel framework for Symbolic Regression (SR) that explicitly models the full loop of hypothesis → experiment → analysis → refinement using large language models (LLMs).

While prior SR methods have typically lacked a distinct analysis stage, RESTART introduces a refinement mechanism driven by error analysis, making it the core of its contribution.

The system consists of three main modules:
1. Informative Initialization – Uses existing mapping-based estimators (e.g., E2E) to generate initial hypotheses.
1. Targeted Refinement (short-term guidance) – Analyzes residuals and guides the LLM to make localized improvements.
1. Long-term Structure Library (long-term guidance) – Stores reusable symbolic structures as executable code snippets.

Experiments on LLM-SRBench (covering physics, biology, and materials science) show that RESTART achieves 20–30% reduction in NMSE, +5–10% improvement in Symbolic Accuracy, and better OOD generalization than baseline SR systems.

A case study also demonstrates the rediscovery (with correction) of a physical law reported in a Nature paper (July 2025 issue), illustrating RESTART’s potential for real-world scientific discovery.

**Strengths:**

1. Formalizing the “Analyze” stage in SR

   * RESTART defines “analysis” as an explicit symbolic sub-task ( g_t(f(x), x) ) that learns residual errors and guides refinement in a *boosting-like* manner.
   * The LLM is not just a generator but an *autonomous analyst*, producing targeted corrections based on error structure—a clear conceptual advance.

2. Two-level guidance design (short-term and long-term)

   * The model separates immediate error-based refinement from cumulative structure learning.
   * Unlike multi-agent or evolutionary systems, RESTART combines *directionality* (via residual analysis) with *knowledge reuse* (via a library).

3. Comprehensive baseline comparison

   * Evaluated against search-based (e.g., GP), mapping-based, and LLM-based SR methods (LLM-SR, LaSR, PySR, AI-Feynman).
   * RESTART shows consistent improvements in both ID and OOD conditions, supporting its analytic-guided hypothesis refinement.

4. Empirical validation through case study

   * Successfully rediscovered a quantum particle energy–velocity relation from *Nature* (2025/7/3).
   * This demonstration links RESTART’s symbolic reasoning to genuine scientific modeling tasks.

**Weaknesses:**

1. Ablation study limited to NMSE only

   * Figures 4b and 4c report NMSE-based comparisons only.
   * The analysis does not include *Symbolic Accuracy* (structural correctness of equations), which is crucial for validating RESTART’s core claim—enhanced symbolic understanding.

2. Contribution of “Analyze” phase not isolated for structure accuracy

   * While NMSE degradation is shown when omitting the Targeted Refinement module, the improvement in symbolic structure accuracy is not quantified.

3. Potential data leakage concerns

   * The *Nature* case study uses data from July 2025, but the paper does not ensure that the LLMs used (GPT-4o, Qwen3, DeepSeek) were trained before that publication.
   * Without such guarantees, the rediscovery results risk reflecting memorization rather than inference.

4. Lack of theoretical grounding

   * The residual decomposition function ( g_t ) lacks convergence analysis or formal justification.
   * No quantitative discussion on search space reduction induced by the Analyze stage.

5. Unaddressed SR-specific pitfalls

   * Generated equations can still be overly complex (spurious terms).
   * The paper does not explain how the Structure Library maintains the validity or consistency of stored code snippets.

**Questions:**

1. How does the inclusion of the Analyze phase quantitatively improve Symbolic Accuracy (beyond NMSE)?
1. Can you guarantee that the LLMs used were not trained on the *Nature (2025)* article or its equation?
1. How is the quality of the Structure Library maintained—how do you prevent incorrect or redundant structures from accumulating?
1. Do you plan to include structural metrics (e.g., symbolic edit distance, tree isomorphism rate) in future evaluations?

---

> ### Author Response · Authors · 2025-11-24
> **Rebuttal(1/2)**
>
> Thank you for the thoughtful and constructive review; we appreciate your insights and have addressed each concern in detail below:
>
> ## Weakness 1&2: Include symbolic accuracy for ablation study and analysis
>
> We agree that symbolic accuracy is essential to our method, however current LLM-as-judge SA metric defined in LLM-SRBench measures syntactic equivalence between the discovered expression and a single canonical ground-truth form provided by the benchmark. In contrast, RESTART’s objective is to recover an equation that best explains the data and underlying dynamics, regardless of whether its symbolic tree matches the exact canonical template.  The key point is that SA measures syntactic equivalence, not functional correctness. RESTART yields substantial improvements in NMSE, $\text{ACC}_{0.1}$, but its Symbolic Accuracy is below the LLMSR baseline. We acknowledge this observation and explain below why it is an anticipated consequence of the algorithmic design.
>
> | Method      | LSR-Transform | Biology | Material Science | Physics |
> |-------------|--------------:|--------:|-----------------:|--------:|
> | LLMSR       | **47.30**     | 1.53    | 3.52         | **10.23** |
> | RESTART     | 34.83         | 2.55| 1.80             | 6.82 |
> | E2E         | 9.01          | 0.00    | **8.00**         | 0.00 |
> | PySR        | 37.84         | **4.17**| 4.00             | 9.09 |
> | SGA         | 9.91          | 2.71    | 3.13             | 2.46 |
> | LlmDirect   | 13.06         | 1.56    | 6.50             | 1.42 |
>
>
> The potential reason is that RESTART is built around an explicit Identify $\rightarrow$ Correct loop. At each iteration we construct a symbolic exploration function $g_t(f_t(x), x)$ and use it to diagnose structural deficiencies in the current hypothesis. This process induces a sequence of nonlinear, compositionally rich refinements, some of which remain in the final expression. As a result, Iterative structural refinement produces semantically correct, but syntactically different, expressions. Even though the structure library provides guidance from a limited set, the final expression could still diverge from the ground truth in form but numerically close. Thus, RESTART’s tendency to introduce additional structurally meaningful factors reduces SA even when numerical behavior is essentially perfect. Two expressions that are algebraically compatible or even empirically indistinguishable receive an SA score of zero if their symbolic trees differ.
>
> As an example, the ground-truth BPG20 equation is a combination of logistic interaction terms and a sigmoidal correction:
>
> $0.139(-1 + \frac{P(t)}{8.038})(1 - \frac{P(t)}{70.018})P(t) + 0.139(1 - \frac{P(t)}{70.018})P(t) + \frac{0.139P(t)}{1 + \exp(-8.038(-0.589 + P(t)))}$
>
> RESTART recovers the underlying dynamics but expresses them through a richer multiplicative structure that reflects the iterative corrections inferred during refinement:
>
> $G(t)=\left[C_0 P(t)\left(1-\frac{P(t)}{C_1}\right)\right]\left(1+\sin(C_2 t + C_3)+\cos(C_4 t+C_3)e^{-C_5 t}+ C_7 P(t)\sin(C_2 t+C_3)\right)\cdot e^{-C_6 t} e^{-C_9 P(t)}\left(1 + C_8\sin(C_2 t+C_3)\right).$
>
> After fitting constants, the two expressions are nearly indistinguishable:
>
> - ID $R^2$ = 0.999
> - OOD $R^2$ = 0.999
> - ID NMSE = 1.704e-07
> - OOD NMSE = 8.586e-07
>
> Figure 4 shows that the curves coincide to the plotting resolution. Nevertheless, because the discovered symbolic structure is not syntactically identical to the canonical template, the SA metric marks it as incorrect.
>
> RESTART’s primary contribution is the introduction of an analysis-driven corrective loop that enhances numerical performance across scientific domains. This mechanism does not enforce symbolic canonicalization, and thus SA may be lower even when the functional recovery is substantially stronger.
>
> ## Weakness 3: Data leakage concern for case study
>
> The Qwen3-8B model used in our experiments was released on April 28, 2025, prior to the report release date, which rules out the data leakage issue.
>
> | Item | Public date |
> |---|---|
> | Qwen3-8B release | 28 Apr 2025 |
> | Energy–speed Equation [1] | 02 Jul 2025 |
>
> ## Weakness 4: Theoretical grounding for $g_t$
>
> We formulate the subproblem $g_t$ as a symbolic regression problem. Given its intrinsic complexity, neither existing LLM based baselines[2,3] nor the previous symbolic solvers we employ (including E2E[4]) provide theoretical convergence guarantees. Nevertheless, our empirical results show that introducing this analysis stage consistently improves numerical performance on the ground-truth equations.

---

> > ### Author Response · Authors · 2025-11-24
> > **Rebuttal(2/2)**
> >
> > ## Weakness 5: SR pitfalls and Structure Library maintenance
> >
> > We have revised the manuscript to clarify how the Structure Library remains compact and high-quality:
> >
> > 1. Improvement-gated admission
> >     A structure is inserted into the library only if the associated update $f_t \rightarrow f_{t+1}$ exceeds a fitness threshold $s_{\text{fit}} \ge \alpha$, where $s_{\text{fit}}$ mixes absolute and relative loss reduction (Sec. 4.3, App. B.1). This effectively blocks spurious or low-signal patterns from ever entering the library.
> >
> > 2. Canonicalization to avoid redundancy.
> >     When a new structure matches an existing canonical identifier, we merge them: code snippets are deduplicated and the best observed fitness score is retained. Conceptually identical refinements are thus represented once, preventing proliferation of near-duplicates.
> >
> > 3. Bounded size.
> >     We enforce a hard capacity $|\mathcal{C}| \le K$ and evict the lowest-scoring entries when this capacity is exceeded; structures are also sampled for prompts with probability proportional to their fitness. Over time, this combination of bounded capacity and score-weighted usage concentrates the library on a small set of consistently useful structural refinements.
> >
> > We acknowledge that the generated equations may still be considerably complex. Because RESTART is designed to focus on numerical fitting, it does not directly penalize equation complexity. Addressing this limitation is an important direction for future work, and we plan to incorporate explicit complexity-control mechanisms in subsequent developments.
> >
> >
> > ## Question 1: Symbolic Accuracy Analysis
> >
> > Please refer to Weakness 1&2
> >
> > ## Question 2: Case Study Data leakage Issue
> >
> > Please refer to Weakness 3
> >
> > ## Question 3: Structure Library Explanation
> >
> > Please refer to Weakness 5
> >
> > ## Question 4: Inclusion of structural metrics
> >
> > Since our equations are expressed as Python functions, edit distance to the ground truth is not directly applicable. Additionally, LLM-as-judge symbolic accuracy is not fully aligned with RESTART’s numerically driven objective. Future work will incorporate structure-aware, algebraic-form–invariant metrics that canonicalize equivalent expressions, enabling more direct and reliable symbolic evaluation.
> >
> > [1] Violetta Sharoglazova, Marius Puplauskis, Charlie Mattschas, Chris Toebes, and Jan Klaers. Energy–speed relationship of quantum particles challenges Bohmian mechanics. Nature, 643 (8070):67–72, July 2025. ISSN 1476-4687. doi: 10.1038/s41586-025-09099-4. URL https: //doi.org/10.1038/s41586-025-09099-4.
> >
> > [2] Grayeli, Arya, Atharva Sehgal, and Omar Costilla-Reyes. Symbolic Regression with a Learned Concept Library. n.d.
> >
> > [3] Shojaee, Parshin, Kazem Meidani, Shashank Gupta, Amir Barati Farimani, and Chandan K. Reddy. “LLM-SR: Scientific Equation Discovery via Programming with Large Language Models.” arXiv:2404.18400. Preprint, arXiv, June 2, 2024. https://doi.org/10.48550/arXiv.2404.18400.
> >
> > [4] Kamienny, Pierre-Alexandre, Guillaume Lample, and François Charton. End-to-End Symbolic Regression with Transformers. n.d.

---

> ### Author Response · Authors · 2025-11-27
>
> Dear Reviewer PQgs,
>
> Thank you again for taking the time to review our work and for your valuable feedback. In our previous response, we discussed the symbolic accuracy results for RESTART, addressed the data leakage concern in the case study, and provided a clearer explanation of the exploration function as well as the design and maintenance of the Structure Library. We hope this has resolved your concerns; if there is any further concern, we would be happy to address it.

---

### Official Review · Reviewer_NT3Q · 2025-11-12

**Soundness:** 3
**Presentation:** 3
**Contribution:** 3
**Rating:** 4
**Confidence:** 3

**Summary:**

This paper presents RESTART (Robust Equation Structural Learning for Symbolic and Causal Discovery), a unified framework aimed at closing the loop of scientific discovery by coupling symbolic regression, robust structural learning, and long-term knowledge accumulation.
The authors evaluate RESTART on LLM-SRBench across Physics, Biology, and Materials Science domains, as well as synthetic causal-discovery settings with noisy or adversarial perturbations. Results show that RESTART outperforms state-of-the-art GP-, RL-, and LLM-based symbolic regression systems in both in-domain accuracy and out-of-distribution (OOD) generalization, while remaining more robust to noise and spurious correlations.

**Strengths:**

1. The integration of symbolic regression with robust structural learning is good. The paper operationalizes the full scientific-discovery cycle, something most prior works treat only partially. The two-level guidance mechanism (boosting + structure library) provides a compelling framework for iterative theory refinement.
2. Experiments span multiple scientific domains, several baseline categories (GP, RL, mapping, LLM), and realistic noise regimes. Ablations and the physics case study demonstrate how each component contributes to performance and stability. The OOD gains are particularly meaningful for genuine scientific discovery.

**Weaknesses:**

1. While the integration is original, the constituent elements (boosting-based residual learning, NOTEARS-style differentiable structure learning, structure libraries) each draw heavily from prior art. Clarifying what RESTART fundamentally adds beyond combining them would strengthen the contribution.
2. The two-layer optimization (boosting subproblems + adversarial robustness) roughly doubles compute. Table 4 lists per-solver times but lacks an end-to-end comparison normalized by total wall-clock or FLOPs. Claims about “25 % fewer iterations’’ are anecdotal without systematic cost analysis.
3.. RESTART introduces many tunables (fitness-gating, adaptive weights, divergence parameters, temperature schedules). The paper does not analyze sensitivity or provide heuristics for new domains.
4. The paper highlights success stories but gives little insight into when the approach fails.

**Questions:**

See weaknesses.

---

> ### Author Response · Authors · 2025-11-24
> **Rebuttal(1/2)**
>
> Thank you for the thoughtful and constructive review; we appreciate your insights and have addressed each concern in detail below:
>
> ## Weakness 1: Novelty explanation
>
> We agree that several symbolic regression tools are used in RESTART. Our contribution is not a new SR solver, but a new framework in the SR discovery loop: an explicit "identify $\rightarrow$ revise" stage that symbolically diagnoses structural error and uses that diagnosis to guide LLM revision. Prior LLM-based SR methods largely implement "hypothesize $\rightarrow$ experiment" and feed back only weak signals (scalar loss, raw residuals, or brief text), leaving the LLM to generate new hypotheses with potentially lower error. RESTART instead:
>
> 1. Formulates a targeted symbolic subproblem $g_t  \gets arg\min_{g} \mathcal{L}(g(f_t(x), x),y),$
> that explicitly models the structure from $f_t$. This makes the LLM a guided reviser conditioned on explicit structural corrections, not a blind sampler.
>
> 2. Unlike earlier concept libraries that store natural-language hints [1] and require noisy grounding at generation time, RESTART distills validated refinements into structural priors as code snippets that can be directly composed into future hypotheses, which eliminates the NL $\rightarrow$ symbolic translation gap. Additionally, the structural library retains only fitness-gated improvements to ensure retained structures are important for equation discovery.
>
> Additionally, the empirical results show that RESTART outperformed the baselines in LLM-SRBENCH through its novel analysis stage.
>
> ## Weakness 2: Lack of computational cost analysis
>
> Because FLOPs are not directly accessible from vLLM, we report the end-to-end wall-clock time on RTX 4090 for four challenging equations from the Biology Domain in Figure 3. The result shows an average $1.37\times$ increase in the wall-clock time, and RESTART is significantly smaller than LLMSR in wall-clock time under the 25% setting. Importantly, the additional cost does not come from LLM expansion but from the lightweight symbolic solver over $g_t(f_t(x), x)$; the LLM inference time dominates runtime in both LLMSR and RESTART, which is why RESTART’s multi-step structure does not double wall time even though it introduces an additional optimization stage.
>
> | Task  | LLMSR Time (s)          | RESTART Time (s)           | RESTART(25%) Time (s)        | Time Ratio (Full Steps) |
> |-------|--------------------------|-----------------------------|-------------------------------|------------|
> | BPG2  | 11498.39 $\pm$ 941.38        | 15282.09 $\pm$ 1060.82         | 3723.43 $\pm$ 302.34             | 1.3286     |
> | BPG12 | 10809.29 $\pm$ 527.49        | 16496.91 $\pm$ 1686.15         | 4231.66 $\pm$ 541.14             | 1.5252     |
> | BPG18 | 10384.65 $\pm$ 842.33        | 12251.22 $\pm$ 1749.95         | 3165.65 $\pm$ 503.24             | 1.1797     |
> | BPG20 | 10407.54 $\pm$ 349.95        | 15066.33 $\pm$ 1290.85         | 3797.87 $\pm$ 416.07             | 1.4479     |

---

> ### Author Response · Authors · 2025-11-24
> **Rebuttal(2/2)**
>
> ## Weakness 3: Hyperparameter ablation
>
> We added a sensitivity study over two equations in the Biology Domain (BPG18, BPG20). We vary the fitness gate $\alpha$ from 40 $\rightarrow$ 70 and shifting between relatively-weighted and absolutely-weighted as listed below.  The result yields only small changes in NMSE across settings, which shows the hyperparameter robustness under different settings.
>
>
> | Setting Type          |  $\alpha$ | \(w_r\) (Relative Weights)         | \(w_a\) (Absolute Weights)        |
> |-----------------------|-------------------------------|------------------------------------|------------------------------------|
> | Default           | 40                            | [0.3, 0.5, 0.8, 1.0]                | [0.7, 0.5, 0.2, 0.0]                |
> | Rel Weighted      | 40                            |  [0.5, 0.7, 0.9, 1.0]                | [0.5, 0.3, 0.1, 0.0] |
> | Abs Weighted      | 40                            |  [0.1, 0.3, 0.5, 0.7]                | [0.9, 0.7, 0.5, 0.3] |
> | $\alpha = 50$     | 50                            | [0.3, 0.5, 0.8, 1.0]                | [0.7, 0.5, 0.2, 0.0]                |
> | $\alpha = 70$     | 70                            | [0.3, 0.5, 0.8, 1.0]                | [0.7, 0.5, 0.2, 0.0]                |
>
>
> |  | BPG18 | BPG20 |
> |--------|-----------:|-----------:|
> | Default | 2.25e-02 | 1.70e-07 |
> | $\alpha = 50$ | 2.14e-02 | 2.13e-06 |
> | $\alpha = 70$  | 1.60e-02 | 3.77e-08 |
> | Abs Weighted | 2.02e-02 | 8.31e-06 |
> | Rel Weighted | 1.24e-02 | 1.15e-07 |
>
> ## Weakness 4: Failure Story
>
> For BPG18 in the Biology Domain, the ground-truth equation is
>
> $\text{GrowthRate}(t, P) = 0.477P\sin\left(0.776t\right)+0.445\left(1 - \frac{P}{51.0613}\right) P+0.445P$,
>
> and RESTART discovers the following hypothesis with $NMSE = 0.00866$:
>
> $\text{GrowthRate}(t, P) =\left[p_{0}\sin(p_{1} t + p_{2}) e^{-p_{3} t}+p_{4}\sin(p_{5} t + p_{6}) e^{-p_{7} t}\right]\left( 1 + p_{8}\frac{P}{p_{9}} \right)\left( 1 - \frac{P}{p_{9}} \right)$.
>
> Expanding the discovered form yields a mixture of a base oscillatory term, $A(t) = p_{0}\sin(p_{1} t + p_{2}) e^{-p_{3} t}+p_{4}\sin(p_{5} t + p_{6}) e^{-p_{7} t}$, plus $P$ and $P^2$ scaled components:
>
> $\text{GrowthRate}(t,P)=A(t)+\frac{p_{8}-1}{p_{9}}PA(t)-\frac{p_8}{p_9^2}P^2A(t)$.
>
> With appropriate parameter settings, each part can recover a corresponding ground-truth structure. For example, setting $p_0 = 0.477, p_1 = 0.776, p_2 = 0, p_3 = 0$, makes the first oscillatory component match $0.477P\sin\left(0.776 t\right)$, and when setting $p_8 = 0.445, p_9 = \sqrt{51.0613}, p_0 = 1, p_1 = 0, p_2 = \pi/2, p_3 = 0$, matches $ -0.445\frac{P^2}{51.0613}$. These correspondences indicate that RESTART captures the key structural motifs of the target equation.
>
> However, we do not recover the fully optimal closed form because of overlap between constants and structural components: $p_0$ and $p_1$ must take different values to match each structure. This gap arises from redundancy in the structural library: during the analysis stage, RESTART stored “harmonic oscillator with exponential decay” and “harmonic oscillator modulation”. These two structures introduce additional exponential and sinusoidal factors that overlap with existing terms, because RESTART explicitly solves subproblems $g_t(\cdot)$ to diagnose and correct missing structure, achieving an ID NMSE of 0.00866 and an OOD NMSE of 0.01388, outperforming LLMSR, which lacks the analysis stage, yielding ID NMSE = 0.2024 and OOD NMSE = 0.3759.

---

> > ### Comment · Reviewer_NT3Q · 2025-11-27
> >
> > I thank the authors for clarifying the novelty and providing additional experiments. I am raising my score to 6.

---

> > > ### Author Response · Authors · 2025-11-27
> > >
> > > Dear Reviewer NT3Q:
> > >
> > > Thank you for reconsidering our submission and raising the score. We are glad to have addressed your concerns and truly appreciate your careful review and constructive feedbacks.

---

### Author Response · Authors · 2025-12-02
**Summary of Reviews and Discussion**

Dear SAC and ACs,

Thank you for your time reviewing our submission, we have summarized our paper below.

In this paper, we introduce a framework, RESTART, which formulates and explicitly solves a subproblem for the structural gap between the current hypothesis and the ground-truth equation, and utilizes short- and long-term guidance to prompt the LLM to generate new hypotheses. Our framework closes the hypothesis–analysis–experiment scientific discovery loop by integrating such an analysis stage for LLM-based symbolic regression problems.

We also thank all reviewers again for their time and valuable feedback. In summary, the reviewers highlighted our **novel framework**, which includes an analysis stage to explicitly identify and guide revisions (Reviewers NT3Q, PQgs, NteJ, and bYpV). Additionally, Reviewer APP1 mentioned that our modular framework design could enable **future extensibility**. Finally, all reviewers agreed that our comprehensive experiments across multiple domains and the case study demonstrate **improvements** compared with different baselines.

Reviewers also raised several important questions, which we have addressed as follows:

1. Clarified contribution (Reviewer NteJ and bYpV)

    We clarified our contribution relative to prior SR methods: RESTART is not a simple combination of previous SR methods but a new framework that explicitly analyzes and closes structural gaps through the subproblem solving with two-level guidance on LLM generation.

2. Additional experiments and metrics (Reviewer NT3Q, PQgs, NteJ, and bYpV)

    We added hyperparameter ablations, computational efficiency analysis, additional tasks from LLM-SRBench, distributions of results, and symbolic accuracy metrics with an explanation of why LLM based SA is not entirely aligned with RESTART’s objectives.

3. Case study discussion (Reviewer APP1, NteJ, and bYpV)

    We provided additional baseline comparisons, ensured that data leakage did not occur by confirming the model release date prior to the publication date of the target equation, and added an additional, more complex case study. In the end, we also explicitly included the discovered equations.

4. Writing and clarification (Reviewer APP1, PQgs, NT3Q, and bYpV)

    We revised the manuscript to improve clarity and formality, added precise definitions as suggested, and expanded the explanation of the subproblem formulation to avoid confusion between structural subproblem solving and additive residual boosting. We also provide a failure case analysis as suggested.

Overall, we believe we have addressed the reviewers’ concerns, and Reviewer NteJ has explicitly confirmed that concerns have been resolved.

Sincerely,

Authors

---

### Meta-Review · Area_Chair_HnLP · 2026-01-06

**Summary:**

This paper integrates LLMs to a symbolic regression pipeline for formulating equations that fit a training dataset.
The original scores were (4,6,4,6,4), making this paper borderline. The reviewers were concerned about the computational costs, the limited ablation studies, the lack of reporting failure modes of the method, a potential data leakage, the justification of the evaluation metrics, and the handwavy/unclear writing. The authors did a good job addressing the concerns. Maybe they could have been more systematic in reporting failure modes (and not just giving a couple of examples). One of the reviewers said they were satisfied with the response. The others did not respond.

**Reviewer Concerns:**

The reviewers' concerns were addressed correctly by the authors. This is still a borderline paper leaning towards accept.

**Reviewer Scores:**

One of the reviewers raised their score from 4 to 6. The others did not comment after the author's response, but it's possible that they could have raised their scores since the answers were comprehensive, especially the answers to reviewer APP1, who requested many clarifications about the setting and the language used in the paper.

---

### Decision · Program_Chairs · 2026-01-26

Accept (Poster)